# miR-143/145 differentially regulate hematopoietic stem and progenitor activity through suppression of canonical TGFβ signaling

Jeffrey Lam[1,2], Marion van den Bosch [1,2], Joanna Wegrzyn[1,2], Jeremy Parker[1], Rawa Ibrahim[1,2], Kate Slowski[1,2], Linda Chang[1,2], Sergio Martinez-Høyer[1,2], Gianluigi Condorelli [3], Mark Boldin[4], Yu Deng[1,2], Patricia Umlandt[1,2], Megan Fuller[1,2] & Aly Karsan[1,2]

Expression of miR-143 and miR-145 is reduced in hematopoietic stem/progenitor cells (HSPCs) of myelodysplastic syndrome patients with a deletion in the long arm of chromosome 5. Here we show that mice lacking miR-143/145 have impaired HSPC activity with depletion of functional hematopoietic stem cells (HSCs), but activation of progenitor cells (HPCs). We identify components of the transforming growth factor β (TGFβ) pathway as key targets of miR-143/145. Enforced expression of the TGFβ adaptor protein and miR-145 target, Disabled-2 (DAB2), recapitulates the HSC defect seen in miR-143/145$^{-/-}$ mice. Despite reduced HSC activity, older miR-143/145$^{-/-}$ and *DAB2*-expressing mice show elevated leukocyte counts associated with increased HPC activity. A subset of mice develop a serially transplantable myeloid malignancy, associated with expansion of HPC. Thus, miR-143/145 play a cell context-dependent role in HSPC function through regulation of TGFβ/DAB2 activation, and loss of these miRNAs creates a preleukemic state.

[1] Michael Smith Genome Sciences Centre, BC Cancer Research Centre, Vancouver, BC V5Z 1L3, Canada. [2] Department of Pathology and Laboratory Medicine, University of British Columbia, Vancouver, BC V6T 2B5, Canada. [3] Department of Cardiovascular Medicine, Humanitas Clinical and Research Center, 20089 Rozzano, MI, Italy. [4] Department of Molecular and Cellular Biology, Beckman Research Institute, City of Hope, Duarte, CA 91010, USA. These authors contributed equally: Jeffrey Lam, Marion van den Bosch. Correspondence and requests for materials should be addressed to A.K. (email: akarsan@bcgsc.ca)

Myelodysplastic syndrome (MDS) is a collection of hematopoietic malignancies in which genomic abnormalities within the CD34$^+$ hematopoietic stem/progenitor cell (HSPC) compartment lead to ineffective hematopoiesis and morphological dysplasia of blood cells[1]. As a result, the primary causes of mortality in MDS patients are consequent to the effects of peripheral blood cytopenias[2]. As well, there is a significantly increased risk of transformation to acute myeloid leukemia (AML)[3]. The most common chromosomal aberration in MDS is an interstitial deletion of chromosome 5q, termed del(5q)[4].

Our lab has previously shown that expression of miR-143, miR-145, and miR-146a are reduced in del(5q) MDS, and that deficiency of miR-145 and miR-146a contributes to thrombocytosis, neutropenia, and megakaryocytic dysplasia, recapitulating some of the features of del(5q) MDS[5]. However, the commonly deleted region (CDR) in del(5q) MDS, which spans 1.5 Mb on band 5q32, only encompasses two of the three microRNAs (miRNAs), miR-143 and miR-145, which are expressed at significantly lower levels in CD34$^+$ cells from del(5q) MDS patients[4,6,7]. When the deletion on chromosome 5 does not encompass the miR-146a locus, its expression is not reduced in patient cells[5]. Given that miR-143 and miR-145 are the only two differentially expressed miRNAs located in the CDR of del(5q) MDS patients, and are expressed as a single primary transcript, we investigated the role of these miRNAs in hematopoiesis and their potential role in del(5q) MDS.

We show that the loss of miR-143/145 reduces HSC activity through ectopic activation of Smad-dependent transforming growth factor-β (TGFβ) signaling. Constitutive activation of TGFβ signaling in HSPC through enforced expression of the miR-145 target, Disabled-2 (DAB2), recapitulates the HSPC defect. In MDS, HSPC can represent up to half the nucleated cells in the bone marrow, compared to 0.5–3% in healthy controls[8]. Similarly, in this mouse model, enforced DAB2 expression in vivo results in hematopoietic progenitor cell (HPC) expansion with time, but continued repression of HSC activity, which in some animals progresses to a myeloid malignancy comprised of expanded HPC. Myeloid expansion is also seen in a proportion of miR-143/145-targeted mice with age, associated with HPC expansion and myeloid infiltration of the liver and spleen, consistent with a MPD. Taken together, our data show that miR-143 and miR-145 are required for HSC maintenance through suppression of Smad-dependent TGFβ/DAB2 signaling. Furthermore, loss of these miRNAs results in differential TGFβ pathway activity in HSPC subpopulations and low but increased risk of leukemic transformation.

## Results

**Loss of miR-143/145 reduces HSC number.** miR-143 and miR-145 are transcribed as a single pri-miRNA transcript[9] and we found that the expression of the two mature miRNAs[10] is strongly correlated in patients with myeloid malignancy (Supplementary Fig. 1a). In contrast, miR-146a expression is not correlated with either miR-143 or miR-145 in the same subset of patients (Supplementary Fig. 1b). Consistent with their localization in the CDR, miR-143 and miR-145 are significantly downregulated in HSPC of del(5q) MDS patients[5,11]. Patients with deletions extending much beyond the CDR on chromosome 5q, and including the miR-146a locus, have more aggressive disease[12,13]. Interestingly, in 59% of low-risk del(5q) MDS, the miR-146a locus is not deleted (Supplementary Table 1). This suggests that the less aggressive form of disease seen in MDS with isolated del(5q) may in part be associated with depletion of miR-143 and miR-145 through a mechanism independent of miR-146a

haploinsufficiency. We thus investigated the role of miR-143 and miR-145 in hematopoietic cells using a gene-targeted mouse model with deletion of Mir143 and Mir145 (miR-143/145$^{-/-}$), as being more representative of lower-risk preleukemic states.

Wild-type (WT), miR-143/145$^{+/-}$, and miR-143/145$^{-/-}$ mice were analyzed for long-term HSC (LT-HSC), short-term HSC (LSK; Lin$^-$Sca1$^+$c-Kit$^+$), common myeloid progenitors (CMPs), granulocyte–macrophage progenitors (GMPs), and megakaryocyte–erythrocyte progenitors (MEPs). At 8–12 weeks, miR-143/145$^{-/-}$ mice showed significantly reduced LT-HSC compared to WT mice ($P = 0.04$), but significant differences were not observed in committed myeloid progenitor populations (Fig. 1a). miR-143/145$^{+/-}$ marrows also showed a trend toward reduced LT-HSC ($P = 0.07$). miR-143/145$^{-/-}$ and miR-143/145$^{+/-}$ mice had similar marrow cellularity to WT mice suggesting a deficit in absolute number of HSC (Fig. 1b).

To examine the effects of miR-143/145 loss on HSPC activity in vitro, we performed clonogenic progenitor (colony-forming unit (CFU)) assays using marrow cells from WT, miR-143/145$^{+/-}$, or miR-143/145$^{-/-}$ mice. Primary progenitor assays did not show significant differences between the total number or types of colonies formed, consistent with the immunophenotypically enumerated progenitor numbers above (Fig. 1c and Supplementary Fig. 1c). Given the pronounced reduction in LT-HSC frequency, we performed serial CFU assays by replating cells from the primary culture. Secondary CFU assays revealed significantly fewer miR-143/145$^{-/-}$ colonies compared to WT ($P = 0.01$) (Fig. 1d), consistent with a defect in more primitive HSPC.

To confirm the HSC defect in miR-143/145$^{-/-}$ mice, we assessed the frequency of functional HSC in vivo in a limiting dilution assay (LDA), by transplanting WT or miR-143/145$^{-/-}$ marrow into WT recipients. Donor engraftment assessed at 16–18 weeks showed that miR-143/145$^{-/-}$ mice had a 2.4-fold reduction in LT-HSC compared with WT mice (1 in 52,793 vs. 1 in 21,664, $P = 0.03$) (Fig. 1e and Supplementary Table 2). HSC activity was not significantly reduced in miR-143/145$^{+/-}$ mice. Taken together, the data indicate that loss of miR-143 and miR-145 reduces the LT-HSC population.

**TGFβ signaling is activated in del(5q) MDS patients.** miRNA are small RNA molecules that are able to negatively regulate multiple target genes. Using an miRNA-target prediction algorithm[14] (TargetScan v6.2), we generated a non-redundant list of miR-143 and miR-145 targets, and organized these into signaling pathways using Ingenuity Pathway Analysis. Several pathways known to have key roles in HSC maintenance and self-renewal were significantly enriched. In particular, the TGFβ signaling pathway[15] was highly ranked ($P = 1.29 \times 10^{-7}$) based on the level of enrichment of TGFβ pathway-associated genes, including ligands and receptors of the TGFβ superfamily, SMAD proteins, and key adaptor proteins (Fig. 2a).

To determine whether the TGFβ pathway is differentially regulated in the context of miR-143/145 deficiency in MDS, we analyzed gene expression data from CD34$^+$ HSPC from del(5q) patients[16], compared to healthy controls from the same study, using Gene Set Enrichment Analysis (GSEA)[17]. We identified TGFβ-related gene sets from the GO:Biological processes group in molecular signatures database in GSEA. Of 239 analyzed gene sets, the 35 gene sets that met the false discovery rate (FDR) cut-off of 0.25 were differentially enriched in del(5q) MDS cells relative to normal controls (Supplementary Table 3). These gene sets were associated with positive regulation of TGFβ signaling (Fig. 2b), transmembrane receptor serine-threonine kinase signaling, and extracellular matrix-related pathways. To identify which genes contributed to the

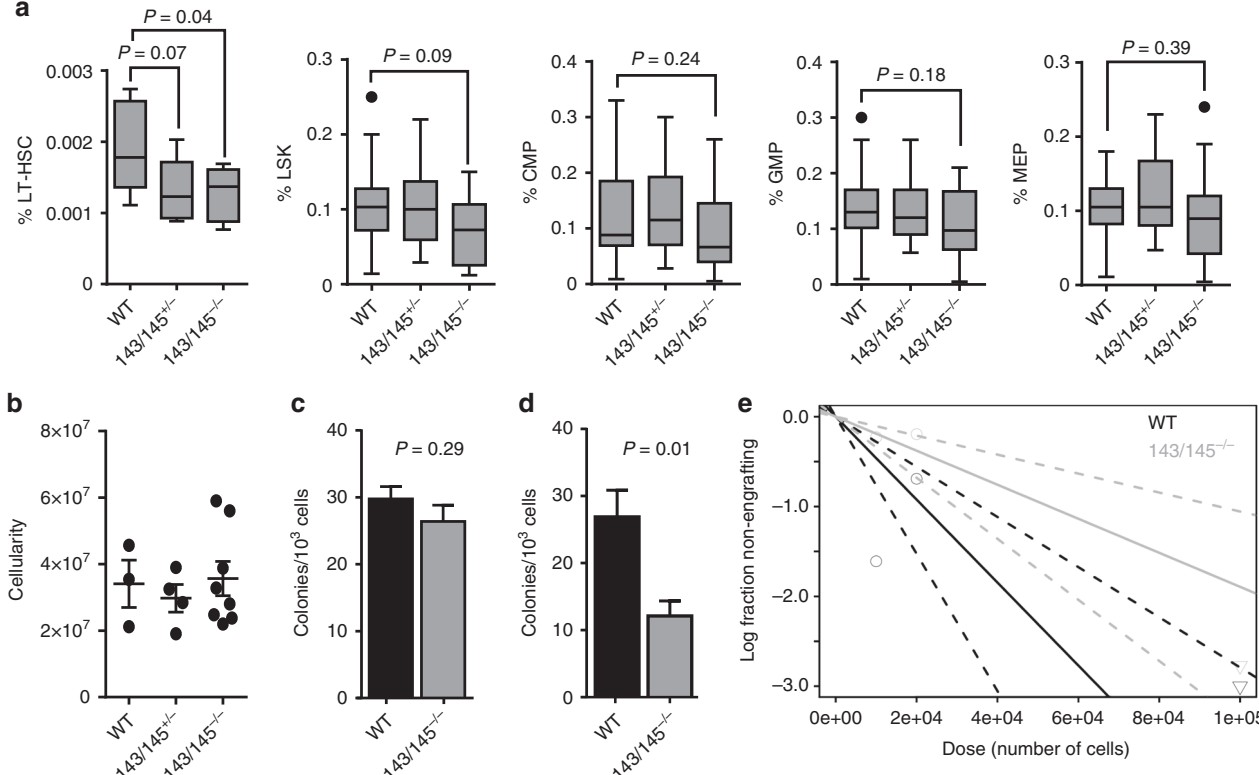

**Fig. 1** Loss of miR-143/145 results in reduced LT-HSC frequency. **a** Frequency of long-term-HSC (LT-HSC, CD45⁺EPCR⁺CD48⁻CD150⁺), LSK/short-term HSC (Lin⁻Sca1⁺c-Kit⁺), common myeloid progenitors (CMP, Lin⁻Sca1⁻c-Kit⁺CD34⁺CD16/32ˡᵒ), granulocyte–macrophage progenitors (GMP, Lin⁻Sca1⁻c-Kit⁺CD34⁺CD16/32ʰⁱ), and megakaryocyte–erythrocyte progenitors (MEPs, Lin⁻Sca1⁻c-Kit⁺CD34⁻CD16/32ˡᵒ) in the marrow of 8–12-week- old wild-type (WT), miR-143/145⁺/⁻, and miR-143/145⁻/⁻ mice, as analyzed by flow cytometry (median ± 1.5 IQR, WT $n = 11$, 143/145⁺/⁻ $n = 9$, 143/145⁻/⁻ $n = 8$). **b** Marrow cellularity (2 femurs and 2 tibias) (WT $n = 3$, 143/145⁺/⁻ $n = 4$, 143/145⁻/⁻ $n = 8$). **c** Colony-forming unit (CFU) assay of marrow cells from WT and miR-143/145⁻/⁻ mice (mean ± SEM, WT $n = 11$, 143/145⁻/⁻ $n = 10$). **d** Primary CFU cells were replated in equal proportions per condition, normalized to the number of input cells, to generate secondary CFUs (mean ± SEM, WT $n = 9$, 143/145⁻/⁻ $n = 7$). **e** Estimate of HSC frequency in WT and miR-143/145⁻/⁻ mice by limiting dilution assay. Shown is a log-fraction plot of the limiting dilution model. The slope of the line is the log-active cell fraction. The dotted lines give the 95% CI

enrichment of those gene sets, we reanalyzed the five most significant gene sets (FDR ≤ 0.05) using leading edge analysis[17] (Supplementary Table 4). The top contributing genes in this analysis are presented in Fig. 2c. *DAB2* was the most differentially expressed TGFβ-related gene predicted to be targeted by both of the miRNAs. DAB2 positively regulates TGFβ signaling by acting as an adaptor that binds the receptor and SMAD proteins, thereby facilitating SMAD2/3 phosphorylation and activation (Fig. 2d)[18].

**miR-143 and miR-145 target Dab2 to regulate TGFβ signaling.** To determine whether increased expression of *DAB2* sensitizes cells to TGFβ pathway activation, we transduced cells with *DAB2* or empty vector followed by transfection of a Smad-responsive luciferase reporter. Following TGFβ stimulation, there was increased reporter activity in *DAB2*-transduced cells compared to control ($P = 0.04$) (Fig. 3a). In line with this, marrow of miR-143/145⁻/⁻ mice showed increased activation of Smad2/3, as determined by intracellular flow cytometry of phospho-Smad2/3, compared to WT marrow following TGFβ stimulation ($P = 0.02$) (Fig. 3b). We then harvested lineage-negative marrow from WT and miR-143/145⁻/⁻ mice to confirm that loss of miR-143 and miR-145 results in increased protein abundance of predicted miR-143/145 targets. Dab2 protein abundance was higher in miR-143/145⁻/⁻ marrow compared to WT marrow ($P = 0.001$)

(Fig. 3c). We also found increased protein abundance of signaling molecules in the TGFβ pathway that act upstream of Dab2: Smad anchor for receptor activation (Sara) and RNA-binding protein with multiple splicing (Rbpms) (Supplementary Fig. 2). In contrast, predicted targets in the Tak1/p38 non-canonical TGFβ pathway were not affected by loss of miR-143/145 (Supplementary Fig. 2), suggesting that miR-143/145 regulates Smad-dependent TGFβ signaling.

To confirm that *DAB2* is also regulated by miR-145 in human cells, we knocked down miR-145 in the human myeloid cell line UT-7 (diploid for chromosome 5q) and observed a corresponding increase in expression of DAB2, and enforced expression of *DAB2* resulted in TGFβ pathway activation as determined by increased phosphorylation of SMAD2/3 (Supplementary Fig. 3a–d). We also observed mRNA induction of TGFβ-dependent genes in human myeloid cells with constitutive expression of *DAB2* (Supplementary Fig. 3e), confirming that derepression of *DAB2* is sufficient to activate the TGFβ pathway.

To demonstrate that *DAB2* is a direct miR-145 target, we inserted the 3′-untranslated region (UTR) of *DAB2* downstream of a luciferase reporter. Co-transfection of reporter and miR-145 constructs resulted in inhibition of reporter activity ($P < 0.0001$) (Fig. 3d). This effect was reversed when all three predicted miR-145 seed sequence-complementary sites were mutated, but not when a single site was mutated in isolation,

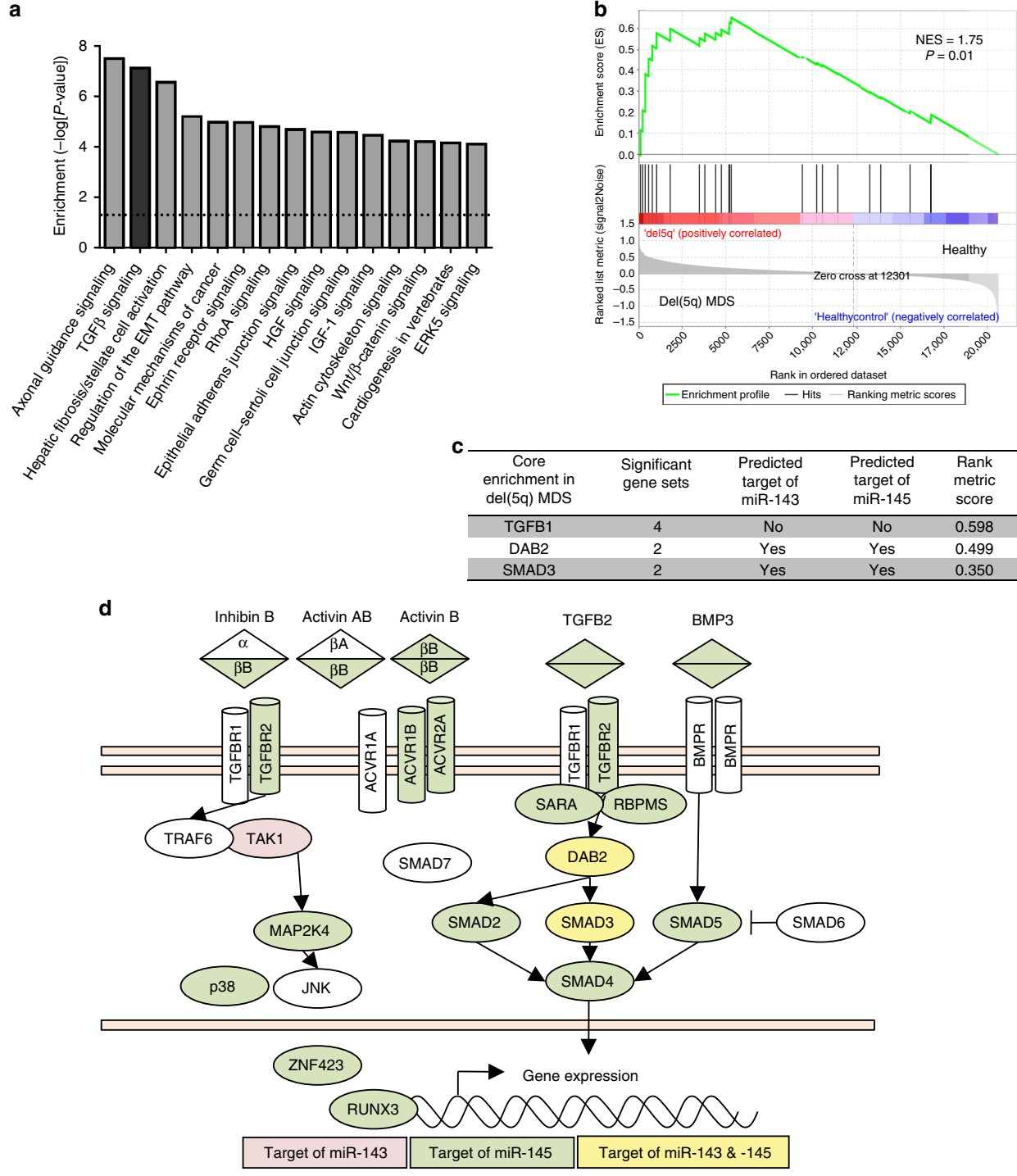

**Fig. 2** miR-143 and miR-145 are predicted to target the TGFβ signaling pathway. **a** Ingenuity pathway analysis of predicted human miR-143/145 target genes curated from Target Scan v6.2. **b** GSEA enrichment plot for the gene set "GO_POSITIVE_REGULATION_OF_CELLULAR_RESPONSE_TO_ TRANSFORMING_GROWTH_FACTOR_BETA_STIMULUS", a top gene set enriched in del(5q) MDS vs. healthy controls, from published gene expression data. **c** Leading edge analysis of the top (FDR ≤ 0.05) five differentially enriched gene sets in del(5q) bone marrow, relative to healthy bone marrow, identifies only three genes present in multiple gene sets, ranked by the GSEA rank metric score. Whether these genes are predicted targets of miR-143 or miR-145 is also indicated. **d** Predicted miR-143/145 target genes in the context of the TGFβ pathway

demonstrating that miR-145 is able to bind the *DAB2* 3′-UTR and inhibit *DAB2* translation through binding of multiple seed-recognition sites (Fig. 3d). Taken together, loss of miR-145 and/or miR-143 in both human and mouse HSPC is sufficient to activate the DAB2/SMAD-dependent TGFβ signaling pathway.

**DAB2 suppresses HSC activity**. To assess the effect of constitutive expression of *DAB2* in mouse marrow HSPC (Supplementary Fig. 4a), we performed clonogenic progenitor assays. DAB2 had a slight effect on progenitor activity in primary CFU assays (*P* = 0.03) (Fig. 4a), and similar to what was observed with miR-143/145$^{-/-}$ marrow, caused greater decrease in colony

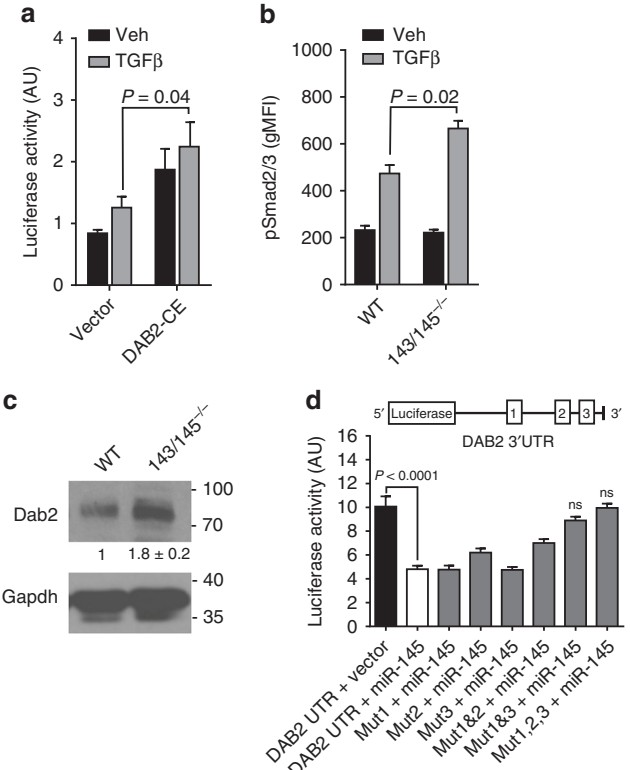

**Fig. 3** Loss of miR-143 and miR-145 activates the TGFβ/Dab2/ Smad3 signaling axis. **a** Measurement of luciferase activity after transfection of 3TP-LUX, a luciferase reporter containing TGFβ-responsive elements, into 293T cells constitutively expressing *DAB2* (DAB2-CE) and stimulated with vehicle (Veh) or 5 ng/ml TGFβ. Data are expressed as arbitrary units (AUs, mean ± SEM, $n = 3$). **b** Phosphorylation of Smad2/3 (pSmad2/3) in lineage-negative marrow cells of wild-type (WT) and miR-143/145$^{-/-}$ mice stimulated with vehicle or 5 ng/ml TGFβ by intracellular flow cytometry. Data are expressed as geometric mean fluorescence intensity (gMFI, mean ± SEM, $n = 3$). **c** Western blot of Dab2 in WT and miR-143/145$^{-/-}$ Lin$^-$ marrow isolated by immunomagnetic negative selection. Densitometry data were normalized to Gapdh and are presented as a ratio relative to vector (mean ± SEM, $n = 7$, $P = 0.001$)). **d** Luciferase activity after co-transfection with empty vector or miR-145 overexpression construct with the WT or seed site-mutated 3′-UTR of *DAB2* inserted downstream of a luciferase reporter (mean ± SEM, $n = 3$)

formation upon serial replating ($P = 0.03$), consistent with DAB2 negatively regulating primitive HSPC (Fig. 4b).

To determine whether constitutive expression of *DAB2* would mimic the defect in miR-143/145$^{-/-}$ HSPC, we transduced WT HSPC with *DAB2*-GFP or Vector-YFP constructs. Cells from both groups were co-transplanted into irradiated recipient mice at a 1:1 ratio in a competitive repopulating experiment (Fig. 4c). Peripheral blood analysis of recipient mice showed a gradual decrease in the proportion of *DAB2*-GFP cells over time (Fig. 4d). Twenty weeks post transplant, analysis of the peripheral blood showed a significantly higher proportion of Vector-YFP cells in mature myeloid (Mac1/Gr1$^+$, $P = 0.03$) and lymphoid (CD19, $P < 0.0001$; CD3, $P < 0.0001$) compartments (Fig. 4e), suggesting reduced repopulating ability of *DAB2*-transduced HSC. We also noted decreased *DAB2*-GFP cells in various myeloid progenitor compartments (CMP, $P = 0.05$; GMP, $P = 0.04$; MEP, $P = 0.03$) (Fig. 4f). However, within the transduced populations (GFP or YFP) there was no difference in the progenitor fractions between the two groups except for GMP ($P = 0.05$) (Fig. 4g). Of interest, in two recipient mice *DAB2*-transduced marrow showed

increased myeloid, but not lymphoid, repopulation compared to the vector control, suggesting a potential for myeloid progenitor accumulation in some cases (Fig. 4e, f). Taken together, the data are consistent with differential effects of *DAB2* on stem and progenitor populations.

To quantify the extent of a potential HSC defect in these mice, we harvested marrow from competitively transplanted mice at 20 weeks post transplantation, and transplanted secondary recipients at limiting dilution (Fig. 4c). After 16 weeks, recipient mice showed 4-fold reduced lymphomyeloid repopulation from *DAB2*-GFP HSCs compared to Vector-YFP HSCs, indicating a defect in functional HSC with constitutive *DAB2* expression ($P = 0.03$) (Fig. 4h and Supplementary Table 5).

**Expression of DAB2 predisposes mice to myeloid malignancy.** To determine the effect of constitutive *DAB2* expression (*DAB2*-CE) in mouse HSPC over time, we transduced HSPC with *DAB2* or Vector constructs and performed three independent transplants using a pool of transduced cells into a total of 18 lethally irradiated recipient mice (Fig. 5a). Mice transplanted with *DAB2*-CE cells showed reduced chimerism compared to Vector cells (42 vs. 65%, $P < 0.01$) (Supplementary Fig. 4b). Peripheral blood immunophenotyping confirmed the reduced contribution of *DAB2*-CE cells to both myeloid (Mac1/Gr1) and lymphoid lineages (CD3/CD19) (Supplementary Fig. 4c) consistent with an HSC defect. Between 25 and 28 weeks post transplant, a subset of *DAB2*-CE mice ($n = 6/18$) started to develop myeloid malignancy consistent with either AML or myeloproliferative disorder (MPD) with increased leukocytes and reduced hemoglobin and platelets, resulting in significantly worse overall survival ($P = 0.01$) (Fig. 5b and Supplementary Fig. 4d). *DAB2*-CE mice that developed AML/MPD had similar long-term engraftment levels to those of *DAB2*-CE mice that did not (47%, $P = 0.5$).

Comparing the geometric mean fluorescence intensity (gMFI) of GFP, a surrogate for the level of ectopic *DAB2* expression, between mice that did and did not develop AML/MPD, we noted a trend towards increased fluorescence in the *DAB2*-AML/MPD group ($P = 0.06$) (Supplementary Fig. 4e). However, there was minimal correlation between endpoint leukocyte counts and GFP$^+$ gMFI in the DAB2-CE mice ($r = 0.32$, $P = 0.09$) (Supplementary Fig. 4f), suggesting that secondary events may cooperate with DAB2 to drive myeloid malignancy.

Moribund mice displayed significantly higher leukocyte counts ($P < 0.0001$), with macrocytic anemia ($P < 0.0001$), thrombocytopenia ($P < 0.0001$), and splenomegaly (Fig. 5c, d). Morphologic examination revealed increased blast cells with some mice displaying frank leukemia and others more closely resembling an MPD (Supplementary Fig. 5a, b). When mice became moribund and were euthanized, marrow cells were sorted based on GFP expression and plated in methylcellulose. In contrast to pre-disease marrows, primary CFU assays showed that GFP$^+$ marrow produced significantly more colonies than GFP$^-$ marrow ($P = 0.01$) (Fig. 5e). To confirm the ability of the leukemic cells to repopulate secondary animals, we performed two independent transplants, into four lethally irradiated recipients each, using unsorted marrow from moribund primary *DAB2*-CE mice with either AML or MPD. Within 4 weeks secondary *DAB2*-CE recipients developed a disease recapitulating the original primary DAB2-AML or MPD (Fig. 5b–d). Moribund AML/MPD mice also had decreased CD19$^+$ B cells and Ter119$^+$ erythroid cells as well as increased Mac1$^+$Gr1$^+$ myeloid cells and primitive CD71$^+$ erythroid cells (Supplementary Fig. 5c).

The paradoxical observation that constitutive *DAB2* expression led to both an HSC defect and leukemic transformation suggested that the leukemic clone in primary *DAB2*-CE AML/MPD mice

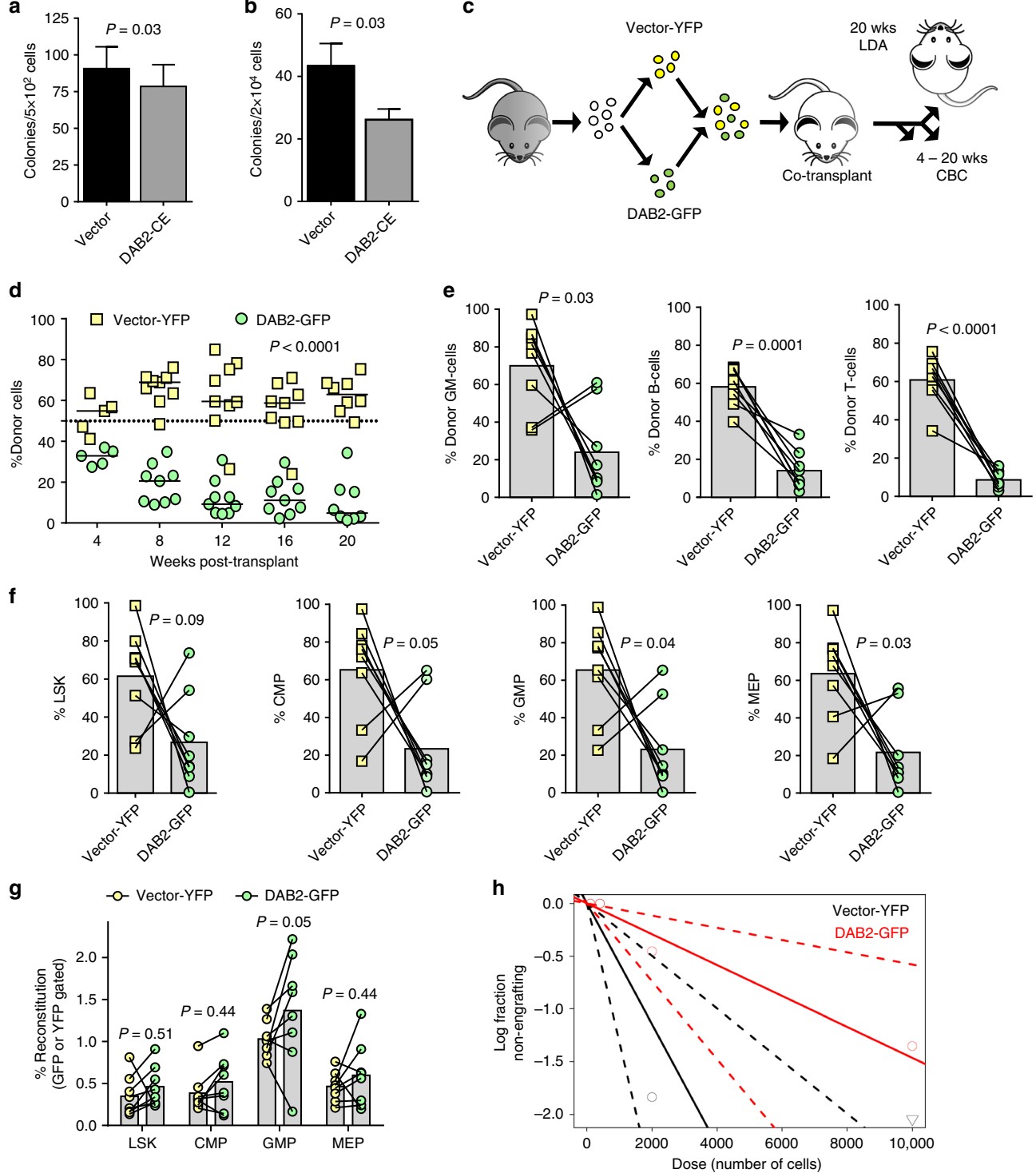

**Fig. 4** DAB2 regulates HSPC function in vivo. **a** Colony-forming unit (CFU) analysis of constitutively expressing *DAB2* (DAB2-CE) marrow (mean ± SEM, *n* = 7). **b** Primary CFU marrow was replated in equal proportions per condition, normalized to the number of input cells, to generate secondary CFUs (mean ± SEM, *n* = 7). **c** Schematic of the transplant experiments shown in **d** through **h**. CBC complete blood counts, LDA limiting dilution assay. **d** Recipient mice were injected with equal numbers of Vector-YFP and *DAB2*-GFP marrow in a competitive assay (mean ± SEM, week 4 *n* = 5, week 8–12 *n* = 9, week 20 *n* = 8). Peripheral blood chimerism was assessed every 4 weeks for 20 weeks. **e** Peripheral blood analysis at 20 weeks (mean, *n* = 8) to assess the percentage of donor granulocyte–monocyte cells (GM; CD45.1+Mac1+ and/or CD45.1+Gr1+), B cells (CD45.1+CD19+) and T cells (CD45.1+CD3+). **f** Bone marrow was harvested at 20 weeks to assess GFP/YFP chimerism (mean, *n* = 8) of LSK (Lin−Sca1+c-Kit+) and myeloid progenitors (CMP, Lin−Sca1−c-Kit+CD34+CD16/32lo; GMP, Lin−Sca1−c-Kit+CD34+CD16/32hi; MEP, Lin−Sca1−c-Kit+CD34−CD16/32lo). **g** Results from **f** were reanalyzed by gating on the transduced population to determine the proportion of specific progenitors within each of the GFP or YFP compartments. **h** Secondary limiting dilution assays were performed using marrow from long-term engrafted mice in **d** (20 weeks post-competitive *DAB2*-GFP/Vector-YFP transplant) and engraftment was evaluated at 16 weeks post-secondary transplant. Shown is a log-fraction plot of the limiting dilution model. The slope of the line is the log-active cell fraction. The dotted lines give the 95% CI

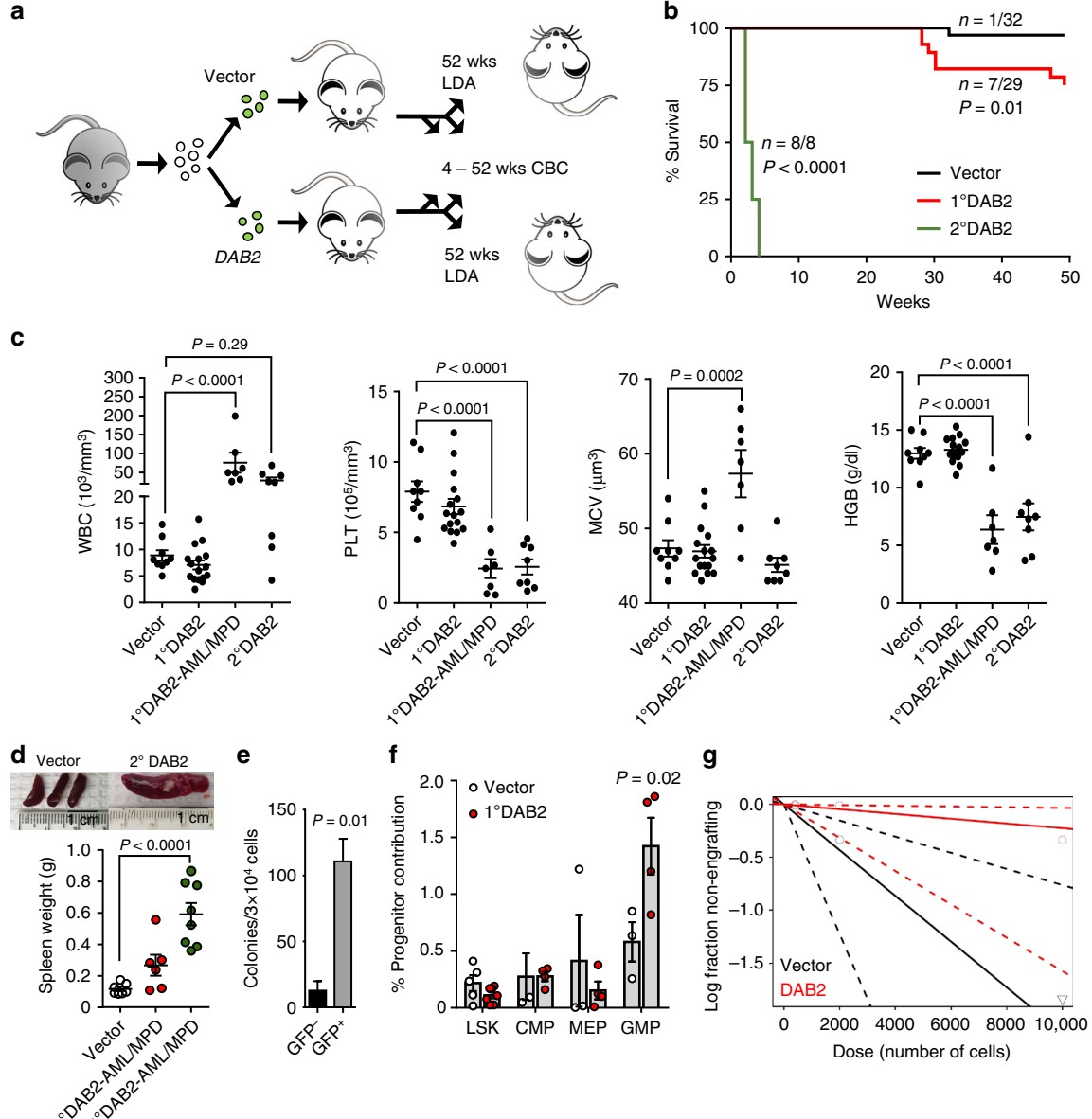

**Fig. 5** Enforced expression of DAB2 predisposes mice to a serially transplantable myeloid malignancy. **a** Schematic of the primary transplant experiment analyzed in **b** through **g**. CBC complete blood counts, LDA limiting dilution assay. **b** A Kaplan–Meier survival curve was generated for mice receiving transplanted marrow containing Vector or *DAB2* constructs following primary (1° DAB2) or secondary transplant (2° DAB2) in those mice dying of myeloproliferation or leukemia. **c** Peripheral blood analysis of white blood cells (WBCs), platelets (Plt), mean cell volume (MCV), and hemoglobin (HGB) at 25 weeks post transplant (mean ± SEM, Vector $n = 9$, 1° DAB2 $n = 16$, 1° DAB2-AML/MPD $n = 7$, 2° DAB2 $n = 8$). **d** Spleen weights were assessed in primary and secondary recipients (mean ± SEM, Vector $n = 6$, 1° DAB2-AML/MPD $n = 6$, 2° DAB2-AML/MPD $n = 8$). **e** CFU analysis was performed using primary *DAB2*-AML/MPD marrow sorted for GFP (mean ± SEM, GFP- BM $n = 3$, GFP + BM $n = 4$). **f** Frequency of LSK (Lin⁻Sca1⁺c-Kit⁺), CMP (Lin⁻Sca1⁻c-Kit⁺CD34⁺CD16/32ˡᵒ), GMP (Lin⁻Sca1⁻c-Kit⁺CD34⁺CD16/32ʰⁱ) and MEP (Lin⁻Sca1⁻c-Kit⁺CD34⁻CD16/32ˡᵒ) determined by flow cytometry in GFP⁺ marrow of 1° DAB2 mice 60 weeks post transplant that did not develop leukemia or a myeloproliferative disorder (mean ± SEM, Vector $n = 3$–5, 1° DAB2 $n = 3$–4). **g** Secondary limiting dilution assays using marrow from long-term engrafted Vector or *DAB2* mice (52 weeks post-primary transplant). Engraftment was evaluated in secondary recipients at 16 weeks post transplant. Shown is a log-fraction plot of the limiting dilution model. The slope of the line is the log-active cell fraction. The dotted lines give the 95% CI

might have originated from a more differentiated progenitor. Indeed, peripheral blood immunophenotyping of primary *DAB2*-CE mice surviving more than 50 weeks post transplant showed an expansion of Mac1⁺Gr1⁺ cells with a significant increase in the GMP compartment of the marrow ($P = 0.02$) (Fig. 5f and Supplementary Fig. 6a, b). These findings are consistent with the observation that *Dab2* expression is inversely correlated with miR-143/145 expression in different stem/progenitor subsets in WT mice, with greatest *Dab2* and lowest miR-145 expression

seen in the GMP population (Supplementary Fig. 7a, b), suggesting that this adaptor supports progenitor expansion. In line with this, in marrow populations of miR-143/145⁻/⁻ mice, *Dab2* expression was higher in progenitors than in more primitive LSK cells (Supplementary Fig. 7c).

To quantify LT-HSC activity in the surviving *DAB2*-CE mice, we performed an LDA 52 weeks post-primary transplant. *DAB2*-transduced marrow showed an 8-fold decrease in LT-HSC frequency compared to vector control ($P = 0.02$) (Fig. 5g and

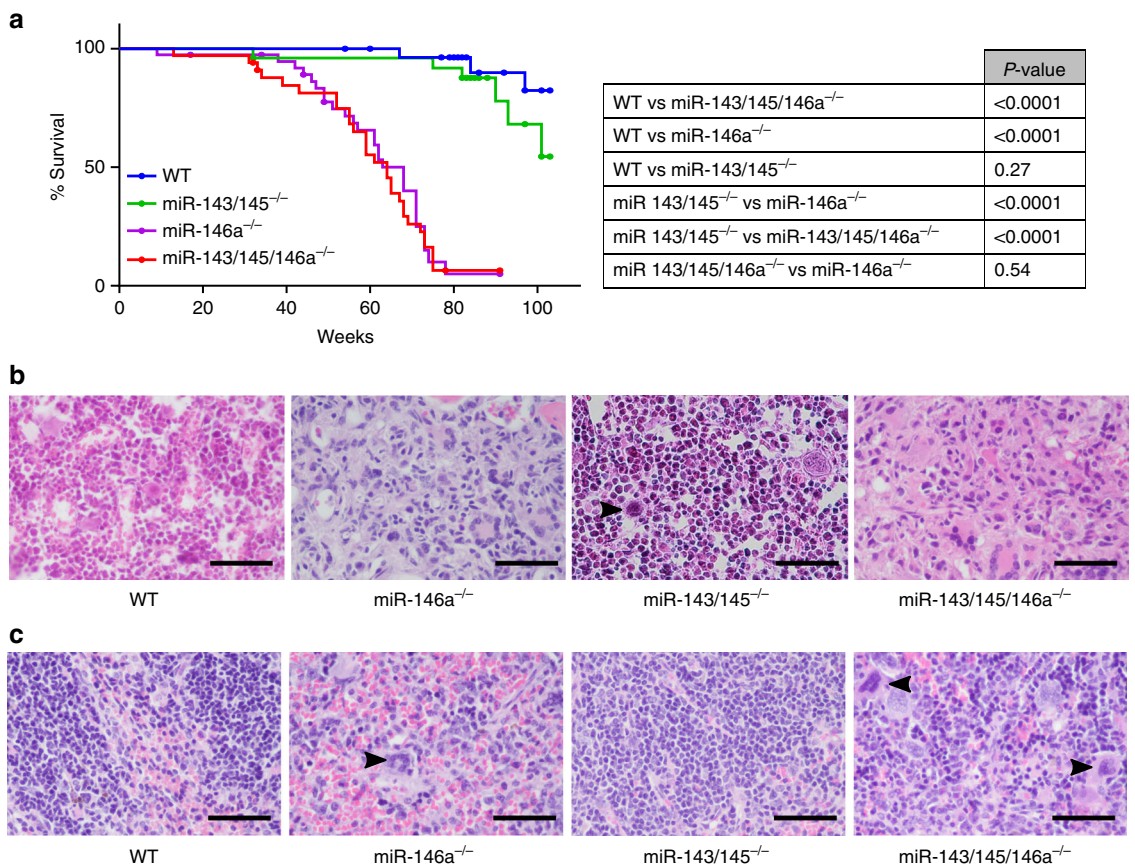

**Fig. 6** Loss of miR-143 and miR-145 results in slight reduction in long-term survival. **a** Kaplan–Meier survival curve of wild-type (WT, $n = 29$), miR-143/145$^{-/-}$ ($n = 20$), miR-146a$^{-/-}$ ($n = 38$), and miR-143/145/146a$^{-/-}$ ($n = 34$) mice. Corresponding $P$ values are indicated in the table. **b, c** Micrographs of H&E-stained marrow (**b**) and spleen (**c**) from WT, miR-146a$^{-/-}$, miR-143/145$^{-/-}$, and miR-143/145/146a$^{-/-}$ mice (scale bar: 50 μm). Hypolobated megakaryocytes (arrow) were distributed through the marrow of miR-143/145$^{-/-}$ mice, while miR-146a$^{-/-}$ and miR-143/145/146a$^{-/-}$ marrows revealed extensive fibrosis. Extramedullary hematopoiesis was evident in miR-146a$^{-/-}$ and miR-143/145/146a$^{-/-}$ spleens (arrows indicate megakaryocytes), while miR-143/145$^{-/-}$ mice showed normal splenic architecture

Supplementary Table 6). Taken together, these data suggest that DAB2 inhibits LT-HSC activity, but in parallel promotes myeloid progenitor activity, and that in some mice secondary events lead to myeloid malignancy that may be initiated in hematopoietic progenitor cells.

**Myeloproliferation in older animals with loss of miR-143/145.** We next examined whether miR-143/145$^{-/-}$ mice would also develop myeloproliferation with time. Although miR-143/145$^{-/-}$ mice appeared to die slightly earlier than WT animals, the difference was not significant. In contrast miR-146a$^{-/-}$ mice showed markedly reduced survival of ($P < 0.0001$) (Fig. 6a). We generated mice deficient for all three miRNAs (miR-143/145/146a$^{-/-}$), which recapitulated deficiency of miR-146a alone[19], including marrow fibrosis and disrupted splenic architecture with extramedullary hematopoiesis, but without obvious additional defects (Fig. 6b, c). These findings suggest that loss of miR-146a is a driver of aggressive disease, but do not explain the impact of miR-143/145 on hematopoiesis and del(5q) MDS.

Given the long latency of a phenotype, we examined a cohort of miR-143/145$^{-/-}$, miR-143/145$^{+/-}$, and WT mice after 1½ years. At 80 weeks, miR-143/145$^{-/-}$ mice showed significantly increased leukocytes ($P = 0.03$) and significantly decreased hemoglobin ($P = 0.05$) and platelets ($P = 0.05$) compared to WT mice (Fig. 7a). This subset of miR-143/145$^{-/-}$ mice also developed hepatosplenomegaly with infiltration of immature myeloid cells in the spleen and liver (Fig. 7b, c and

Supplementary Fig. 8), which was not observed in the WT cohort ($P = 0.05$, Fisher's exact test). Interestingly, white blood cell (WBC) count and spleen/liver weight were positively correlated in miR-143/145$^{-/-}$ mice ($r = 0.53$, $P = 0.001$; $r = 0.56$, $P = 0.001$; respectively) (Fig. 7d, e). There was also a strong positive correlation between spleen and liver weights ($r = 0.93$, $P < 0.0001$) (Fig. 7f). Interestingly, we saw a correlation between WBC count and myeloid progenitor numbers in old miR-143/145$^{-/-}$ mice ($r = 0.82$, $P = 0.02$) (Fig. 7g). Therefore, similar to enforced *DAB2* expression, loss of miR-143/145 reduces HSC activity while increasing progenitor activity leading to myeloproliferation in a subset of mice.

**Loss of miR-143/145 regulates HSPC through TGFβ signaling.** We next tested the functional requirement of Smad3 and Dab2 for hematopoiesis in miR-143/145$^{-/-}$ marrow. We inhibited TGFβ signaling by knocking down *Smad3* or *Dab2* in miR-143/145$^{-/-}$ HSPC (Supplementary Fig. 9a, b) and performed CFU assays. In primary assays, knockdown of *Smad3* or *Dab2* inhibited progenitor activity in WT and miR-143/145$^{-/-}$ cells (Fig. 8a, c), suggesting a positive regulatory effect of TGFβ signaling on progenitor cells, consistent with our previous data above. In secondary replating assays miR-143/145$^{-/-}$ cells transduced with the empty vector formed significantly fewer colonies than WT cells (Fig. 8b, d), similar to parental cells as observed in Fig. 2d. Knockdown of either *Smad3* ($P = 0.002$) or *Dab2* ($P = 0.01$) in miR-143/145$^{-/-}$ marrow rescued hematopoietic

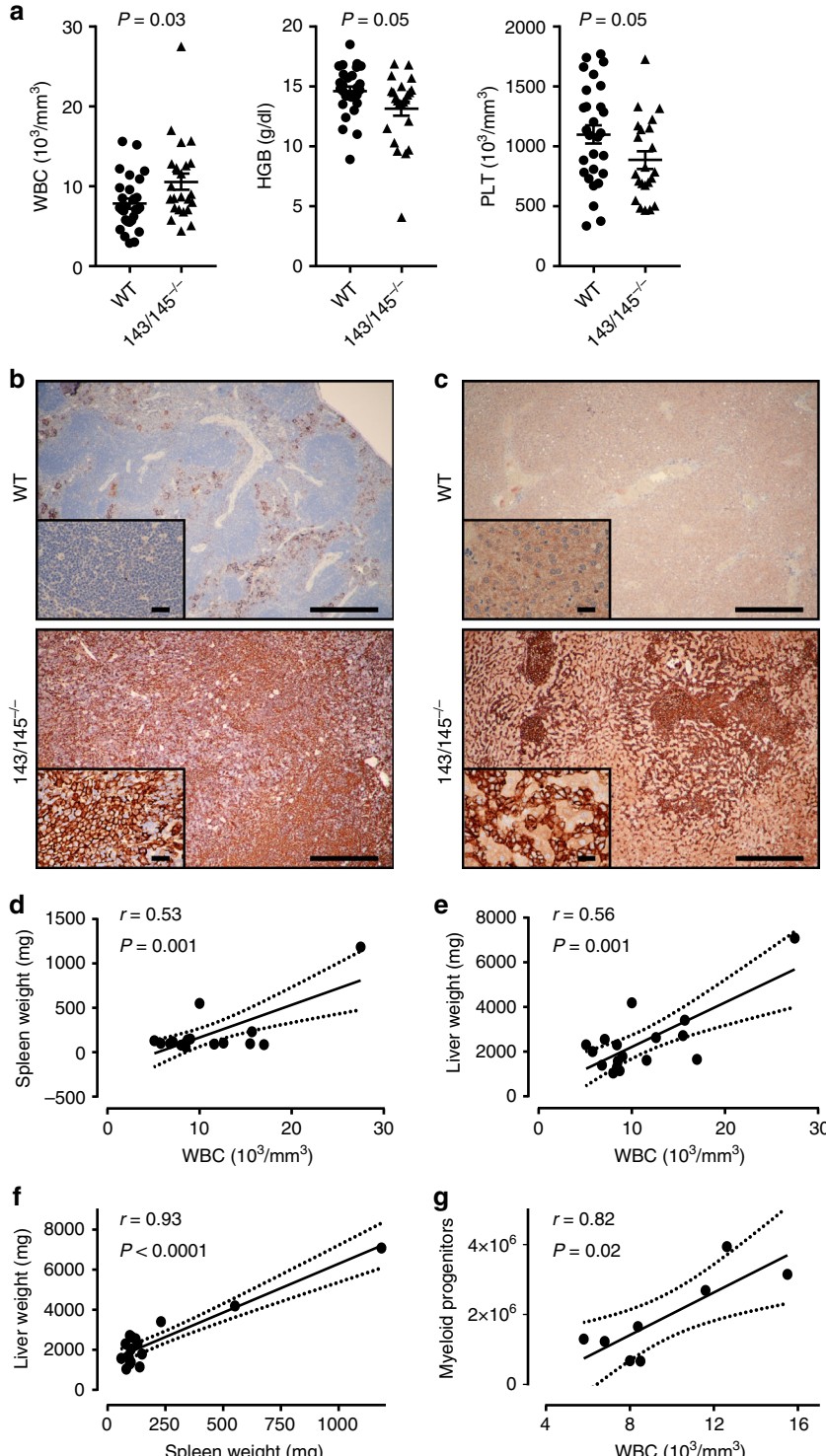

**Fig. 7** Long-term miR-143-145 deficiency results in myeloid expansion. **a** White blood cell (WBC), hemoglobin (HGB), and platelet (PLT) counts in peripheral blood of 80-week-old wild-type (WT; $n = 28$) and miR-143/145$^{-/-}$ mice ($n = 23$). **b** Micrographs of spleen sections from WT and miR-143/145$^{-/-}$ mice immunostained for Mac1 (CD11b) (scale bar: 50 μm). Insets show higher magnification of selected area (scale bar: 100 μm). **c** Microscopic images of liver sections from WT and miR-143/145$^{-/-}$ mice immunostained for CD11b (scale bar: 50 μm). Insets show higher magnification of selected area (scale bar: 100 μm). **d–g** Linear regression and Pearson's correlation analysis was performed on data from aged miR-143/145$^{-/-}$ mice (82–101 weeks old) to compare the relationship between spleen/liver weights and WBC count (**d**, **e**, $n = 17$), spleen and liver weight (**f**, $n = 17$), and WBC count and the number of progenitors (Lin$^-$Sca1$^-$c-Kit$^+$) (**g**, $n = 8$).

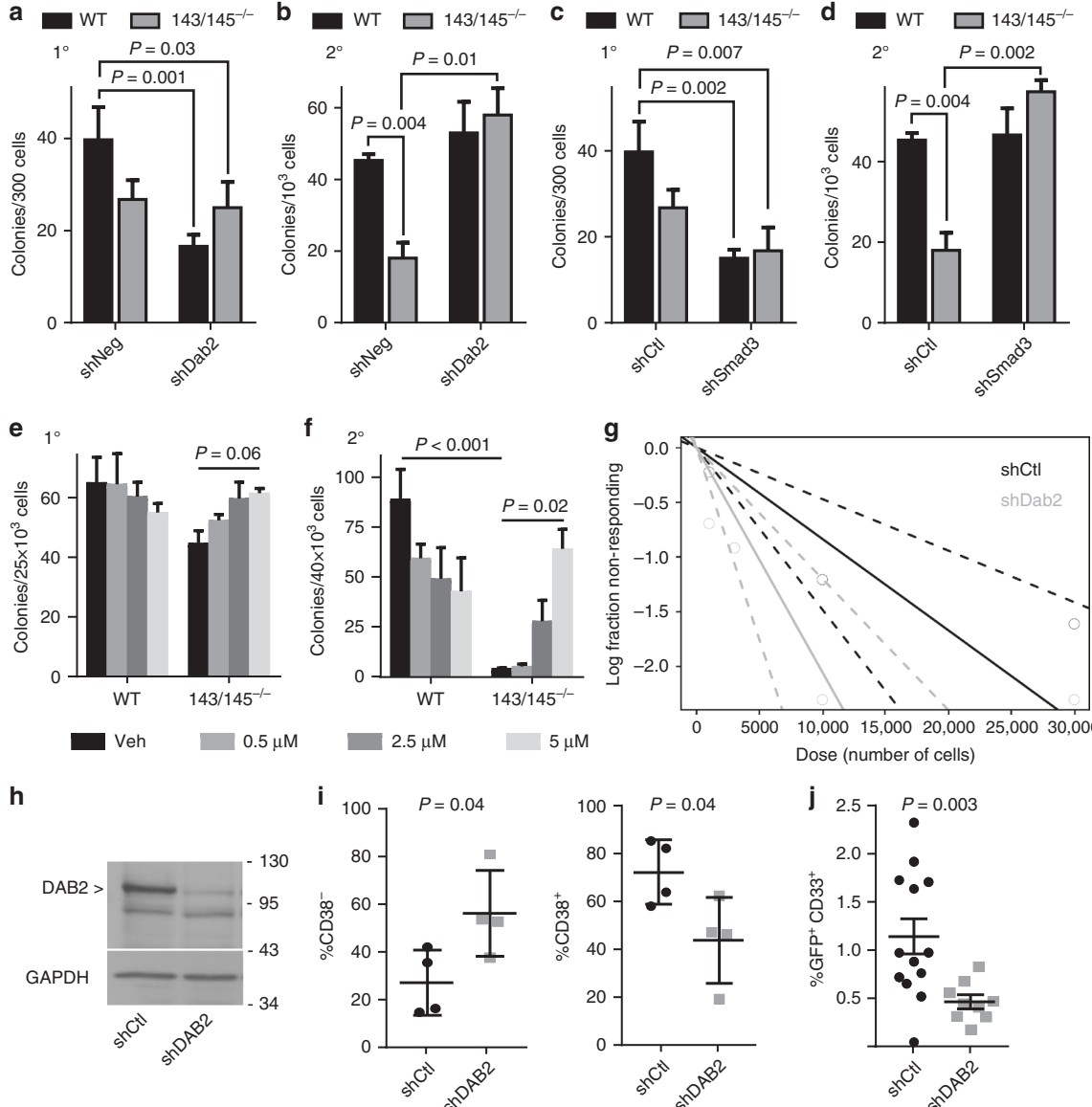

**Fig. 8** Inhibition of TGFβ signaling regulates HSPC in miR-143/145$^{-/-}$ mouse marrow and in human del(5q) MDS cells. **a** Colony-forming unit (CFU) analysis of marrow from WT and miR-143/145$^{-/-}$ mice transduced with shCtl or sh*Dab2* (mean ± SEM, $n = 3$). **b** Primary CFU marrow was replated in equal proportions per condition, normalized to the number of input cells, to generate secondary CFUs (mean ± SEM, $n = 3$). **c** Primary and (**d**) secondary CFU analysis of marrow from WT and miR-143/145$^{-/-}$ mice transduced with shCtl or sh*Smad3* (mean ± SEM, $n = 3$). **e** CFU assay of marrow cells from WT and miR-143/145$^{-/-}$ mice treated with the indicated concentrations of galunisertib (mean ± SEM, $n = 3$). **f** Primary CFU cells were replated in equal proportions per condition, normalized to the number of input cells, to generate secondary CFUs (mean ± SEM, $n = 3$). **g** Marrow was harvested from mice transplanted with miR-143/145$^{-/-}$ marrow transduced with shCtl or sh*Dab2*, and a long-term culture-initiating cell (LTC-IC) assay was performed. Shown is a log-fraction plot of the limiting dilution model. The slope of the line is the log-active cell fraction. The dotted lines give the 95% CI. **h** Western blot of DAB2 to verify protein knockdown in MDS-L cells using shRNA constructs. **i** Flow cytometry analysis of marrow from moribund mice transplanted with del (5q) MDS-L cells transduced with shCtl or sh*DAB2*. The proportion of CD38$^-$ or CD38$^+$ cells within the CD34$^+$ compartment is shown (mean ± SEM, $n = 4$). **j** Peripheral blood from mice transplanted with del(5q) MDS-L cells transduced with shCtl ($n = 13$) or sh*DAB2* ($n = 9$) was analyzed by flow cytometry 5 weeks post transplant. The percentage of GFP$^+$CD33$^+$ cells in the live cell population is shown (mean ± SEM)

progenitor activity upon serial replating, suggesting an inhibitory effect of Smad-dependent TGFβ signaling on more primitive HSPC (Fig. 8b, d). CFU assays were also performed using the SMAD3 small-molecule inhibitor, SIS3[20], which led to similar results (Supplementary Fig. 9c, d). In contrast, inhibition of SMAD-independent signaling, using Tak1 or p38 inhibitors, did not have a differential effect between WT and miR-143/145$^{-/-}$ cells (Supplementary Fig. 9e–h), confirming that the miR-143/145$^{-/-}$ hematopoietic phenotype is driven by Smad-dependent TGFβ

signaling, consistent with the lack of derepression of Smad-independent pathway molecules by miR-143/145$^{-/-}$ deficiency (Supplementary Fig. 2).

To investigate how the level of TGFβ pathway activation affects HSPC function, we performed CFU assays using various doses of the TGFβR1 kinase inhibitor galunisertib (LY2157299), which resulted in multiple degrees of TGFβ pathway inhibition, as measured by phosphorylation of SMAD2/3 (Supplementary Fig. 9i). Greater TGFβ pathway inhibition resulted in greater

functional recovery of miR-143/145$^{-/-}$ marrow upon serial replating ($P = 0.02$), implying a dose-dependent effect of TGFβ pathway activation on more primitive cells (Fig. 8e, f).

To further strengthen the link between TGFβ pathway activation and miR-143/145$^{-/-}$ HSPC function, we transplanted miR-143/145$^{-/-}$ cells transduced with shCtl or sh*Dab2*. Eight weeks post transplant, marrow from the mice was harvested to allow ex vivo analysis of primitive HSPC at limiting dilution in a long-term culture-initiating cell (LTC-IC) assay. HSC frequency was increased in miR-143/145$^{-/-}$ cells with *Dab2* knockdown by approximately 2-fold ($P = 0.01$) (Fig. 8g and Supplementary Table 7), which is similar to the reduction in HSC activity in miR-143/145$^{-/-}$ marrow (Fig. 1e). Together, these data reinforce the proposition that miR-143/145 deficiency results in HSPC that are hyperresponsive to Dab2/Smad-dependent TGFβ signaling, which results in inhibition of primitive HSPC, but activation of more differentiated progenitors.

The human del(5q) MDS cell line, MDS-L, is haploinsufficient for miR-143/145 and is morphologically and immunophenotypically heterogeneous[5,21,22]. Xenografting MDS-L cells into immunodeficient mice results in immunophenotypic differentiation as noted by differential expression of the HSPC markers CD34 and CD38[21]. We thus knocked down *DAB2* in MDS-L cells (Fig. 8h) and xenografted immunodeficient mice. Mice were killed when they became moribund and MDS-L cells from the marrow were examined by flow cytometry. The proportion of more primitive HSPC (CD38$^-$) in the CD34$^+$ cell fraction was increased in sh*DAB2* MDS-L-transplanted mice compared to shCtl ($P = 0.04$), whereas the proportion of more mature HSPC (CD38$^+$) was decreased ($P = 0.04$) (Fig. 8i). In addition, knockdown of *DAB2* significantly reduced engraftment at 5 weeks, as measured by the number of human CD33$^+$ myeloid cells in the blood (Fig. 8j). Thus, in a human del(5q) MDS line, *DAB2* knockdown mimics the effects seen in miR-143/145$^{-/-}$ HSPC, suggesting that derepression of DAB2 and TGFβ signaling through depletion of miR-143 and miR-145 in del(5q) MDS may drive HPC expansion while simultaneously suppressing HSC.

## Discussion

The CDR of chromosome 5q contains miR-143 and miR-145, which are transcribed as a single primary transcript, and show reduced expression in del(5q) MDS[5]. While miR-145 has been shown to target *FLI1*[23] and is responsible for the characteristic megakaryocytic dysplasia[5,11], little else is known about whether and how these two miRNAs regulate hematopoiesis. In this report, we describe the hematopoietic defects in a mouse model targeted for miR-143/145, and demonstrate that loss of these miRNAs results in depletion of LT-HSC. Paradoxically, with age a subset of miR-143/145$^{-/-}$ mice develop myeloproliferation with hepatosplenomegaly, anemia, and thrombocytopenia. Our data point to the hematopoietic effects of miR-143/145 depletion being transmitted mainly through Smad-dependent activation of TGFβ signaling facilitated by derepression of the adaptor protein, Dab2[18].

The effect of TGFβ signaling on the hematopoietic system is complex and varies greatly depending on the context and cell type[15]. Mouse models that target key TGFβ signaling molecules have pointed to a vital role of this pathway in both developmental and adult hematopoiesis. Modulation of TGFβ signaling is being explored for therapeutic purposes in MDS, and TGFβ pathway inhibition has been shown to reverse cell cycle arrest[24–26]. Here, we describe a direct link between the loss of miR-143/145 and the activation of TGFβ signaling in del(5q) MDS. We show that loss of miR-143 and miR-145 results in enhanced activation of the TGFβ pathway in vitro and in vivo. Inhibition of the TGFβ

pathway by interference with *Dab2* or *Smad3* in miR-143/145$^{-/-}$ marrow inhibits activity of more mature progenitors while concomitantly inhibiting more primitive HSPC. Hence, the defect observed in miR-143/145$^{-/-}$ marrow cells is specifically attributable to activation of the TGFβ pathway mediated by Dab2 in a Smad-dependent manner.

The TGFβ adaptor protein, Dab2, is an important target of miR-145 responsible, at least in part, for the phenotype seen in miR-143/145$^{-/-}$ mice. Constitutive expression of *DAB2* in mouse HSPC phenocopied findings in miR-143/145$^{-/-}$ HSPC, including the LT-HSC defect. Although constitutively expressing *DAB2* HSPCs were able to engraft and repopulate recipient mice, *DAB2* expression led to a decrease in LT-HSC. Of interest, differential RNA expression analysis revealed an enriched TGFβ signature in del(5q) MDS patients, suggesting that TGFβ activation may be clinically relevant in del(5q) MDS. Thus, it is possible that *DAB2* overexpression cooperates with other genetic aberrations in MDS to drive the disease phenotype. Given that inhibitors of the TFGβ pathway are currently being trialed in low-risk MDS[27–29], it may be that *DAB2* expression could serve as a biomarker in clinical trials.

Despite a clear defect in HSC maintenance, a subset of mice with constitutive marrow *DAB2* expression developed a transplantable myeloid malignancy at a median of 34 weeks post transplant with a penetrance of approximately 30%. Secondary recipients showed a rapid expansion of Mac1$^+$/Gr1$^+$ cells in the marrow and spleen, associated with CD71 expression, a marker associated with poorly differentiated AMLs[30]. This phenotype echoes what was observed in older miR-143/145$^{-/-}$ mice with leukocytosis, anemia, thrombocytopenia, and hepatosplenomegaly.

As it has proven difficult to xenograft primary MDS cells, HSC function has not been well studied in MDS. However, there is accumulating evidence that MDS initiates in a normal HSC[31–33]. Paradoxically, although activation of the DAB2/Smad-dependent TGFβ pathway inhibited LT-HSC activity early, there was expansion of myeloid progenitor cells concomitant with progression to myeloid malignancy. This finding is remarkably similar to what has been described in MDS, where immunophenotypic HSC numbers show wide variation, but with a very large proportion of patients having low HSC, and significant numbers having low GMP[34]. However, high-risk MDS patients have significant expansion of the GMP-like population[34,35]. Expansion of the myeloid progenitor compartment in MDS samples with higher risk of leukemic transformation is consistent with various reports showing that AML is characterized by an expansion of immature myeloid cells that can originate from GMP-like stem cells, both in mouse models and human disease[36–38]. Indeed a recent report has suggested that transit through a GMP-like stage is a requirement for generating leukemic stem cells in mouse models of AML[39]. Thus, our findings are consistent with an HSC defect, but also indicate that progression of disease is associated with expansion of the myeloid progenitor population, making the current model a good mimic of human MDS, and providing an explanation for the clonal outgrowth of malignant cells in the face of defective HSC function.

We directly demonstrate a functional role of the TGFβ/DAB2 signaling axis in human del(5q) HSPC regulation in vivo using an MDS-L xenograft model. Xenografted MDS-L cells express multiple HSPC markers including CD34 and CD38[21]. Knockdown of *DAB2* causes an increase in more primitive CD34$^+$CD38$^-$ cells and a corresponding decrease in more differentiated CD34$^+$CD38$^+$ progenitors. Our findings imply that constitutive *DAB2* expression drives progenitor expansion in del(5q) MDS, but potentially also in other MDS subtypes where the protein is overexpressed.

The current study provides direct evidence that miR-143 and miR-145 are able to affect the hematopoietic system at the stem cell level through the TGFβ pathway and points to a previously unappreciated role for DAB2 in HSPC function. The defect in HSC function likely contributes to the bone marrow failure phenotype in del(5q) MDS while in parallel driving expansion of the HPC population. In addition to the miRNAs studied here, several coding genes have been implicated in defining the phenotype of del(5q) MDS. Of these, haploinsufficiency of *RPS14* has recently been shown in a mouse model to be responsible for the macrocytic anemia characterizing the disease through induction of the alarmins S100a8 and S100a9, leading to a p53-dependent erythroid differentiation defect[40]. It will be of interest to determine whether the combination of *RPS14* and miR-143/145 haploinsufficiency recapitulates the phenotype of del(5q) MDS.

## Methods

**Mice**. The miR-143/145-targeted and miR-146a-targeted mice were generated previously and maintained on a C57Bl/6J:Pep3b-Ly5.1 (Pep3b) background[19,41]. The two strains were backcrossed to generate mice triply deficient for all three miRNAs. Analysis was done on both male and female mice. For DAB2 transplant studies, marrow from male Pep3b mice was used. Male C57Bl/6J-TyrC2J (C2J) mice were used as recipients (Ly5.2) for mouse cells. For xenotransplants, female NRG-3GS mice were used as recipients. When all mice were of the same age, experimental and control cages were randomly assigned. When ages were different (but within the range specified), the cages were distributed such that neither the experimental nor the control group would be skewed towards older or younger mice. Sick reports were generated by technicians within the animal facility and technicians were not informed which cages/mice were experimental or control. Group size was determined based on data from pilot studies. All strains were bred and maintained in-house at the BC Cancer Research Centre Animal Resource Centre. All animal protocols were approved by the Animal Care Committee of the University of British Columbia (Vancouver, BC, Canada).

**Cell lines and miRNA decoy**. The megakaryocytic leukemia cell line, UT-7 (not present in Database of Cross-Contaminated or Misidentified Cell Lines), was obtained from Deutsche Sammlung von Mikroorganismen und Zellkulturen (DSMZ) and authenticated by short tandem repeat profiling (Genetica DNA Laboratories, Burlington, NC, USA). Cells were maintained in Dulbecco's modified Eagle's medium (DMEM) media supplemented with 15% fetal bovine serum (FBS) and human granulocyte–macrophage colony-stimulating factor. Mycoplasma testing was performed every 2 weeks to ensure the absence of contamination. We cloned eight tandem repeats of miR-145 decoy sequences into the 3′-UTR of YFP and subsequently cloned this fragment into a pLL3.7 lentiviral vector. As previously described[5], each repeat contained a short mismatch sequence (-CTT) at each miRNA binding site at positions 9–12 of the mature miRNA sequence and the repeats are separated by a random spacer sequence. The 293T cells were purchased from ATCC (American Type Culture Collection, USA) and were transfected with the lentiviral construct along with viral packaging vectors VSVG, RRE, and REV. UT-7 cells were cultured in the collected viral supernatant and FACS-sorted for YFP. The MDS-L cell line was obtained from K. Tohyama (Kawasaki Medical School, Japan).

**Luciferase reporter assay**. The DAB2 3′-UTR (1.7 kb) was inserted downstream of a luciferase reporter in the pSGG vector (Switchgear Genomics), while the hsa-mir-145 was cloned into a phosphoglycerate promoter-driven lentiviral vector (pGK)[5]. The predicted seed sites of DAB2 UTR were mutated using the Phusion Site-Directed Mutagenesis Kit. Primers are listed as follows: hDab2UTR_mutseed1F, CCTTTATCAAGTATGACCAAAACTTTTCTTGC; hDab2UTR_mutseed1R, GCAAGAAAAGTTTTGGTCATACTTGATAAAGC; hDab2UTR_mutseed2F, CCTAAATGTGCTAATGACCAGGTAACTATTTCT; hDab2UTR_mutseed2R, AGAAATAGTTACCTGGTCATTAGCACATTTAGG; hDab2UTR_mutseed3F, ACTCATTTCAATAATGACCAGTGGCAGAATATC; hDab2UTR_mutseed3R, GATATTCTGCCACTGGTCATTATTGAAATGAGT. For the DAB2 luciferase assay, 100 ng of pSGG, pSGG-DAB2 UTR, or DAB2 UTRMut DNA was transfected into 293T cells (24-well format) along with 5 ng of pRL-TK renilla and 150 ng of pGK-Empty or pGK-mir-145, using TransIT transfection reagent (Mirus). Where indicated, cells were stimulated with vehicle (4 mM HCl, 1 mg/ml bovine serum albumin (BSA)) or 5 ng/ml TGFβ. Lysates were collected 48 h later using the Promega Luciferase Kit (E1960) and read on a luminometer according to the manufacturer's protocol.

**Western blotting**. Cells were pelleted and lysed in RIPA buffer (25 mM Tris, 150 mM NaCl, 1% NP40, 0.5% sodium deoxycholate, 0.1% sodium dodecyl sulfate (SDS)), to which protease and phosphatase inhibitors were added. Cell extracts were centrifuged at $10,000 \times g$ at 4 °C, and the supernatant was mixed with NuPAGE LDS sample buffer (Invitrogen) which was supplemented with 50 mM dithiothreitol. Samples containing 20–50 μg protein were separated by SDS-polyacrylamide gel electrophoresis and transferred at 100 V for 1 h to nitrocellulose membranes, which were subsequently blocked using 5% milk or BSA in Tris-buffered saline with Tween (20 mM Tris, 137 mM NaCl, 0.1% Tween). Blots were probed with primary antibodies against DAB2 (610464; BD Biosciences, 1:500), SMAD2 (5339; Cell Signaling, 1:500), SMAD3 (9523; Cell Signaling, 1:500), SMAD4 (38454; Cell Signaling, 1:500), SMAD5 (12534; Cell Signaling, 1:500), TAK1 (5206; Cell Signaling, 1:500), MAP2K4 (Santa Cruz Biotechnology; 376838, 1:1000), p38 (9212; Cell Signaling, 1:500), TGFβ RII (R&D Systems; AF532, 1:400), RBPMS (Santa Cruz Biotechnology; 293285, 1:400), SARA (LifeSpan Biosciences; C410575, 1:400), and GAPDH (G8795; Sigma, 1:20000). Subsequently, blots were incubated with horseradish peroxidase-conjugated secondary antibodies. Proteins were detected using ECL reagents. Uncropped western blots are provided in Supplementary Fig. 10.

**Peripheral blood and bone marrow collection and analysis**. Peripheral blood was collected from the tail vein every 4 weeks. Complete blood counts were obtained using a Scil Vet ABC Hematology Analyzer (Scil Animal Care Company). Marrow was harvested from sacrificed mice by flushing the tibiae, femurs, and pelvic bones. After red blood cell lysis, using ammonium chloride lysis solution (0.8% NH₄Cl with 0.1 mM EDTA, STEMCELL Technologies), cells were washed with phosphate-buffered saline (PBS) containing 2% FBS, followed by blocking with 10% rat serum (Sigma). Where indicated, cells were stimulated with vehicle (4 mM HCl, 1 mg/ml BSA) or 5 ng/ml TGFβ. In case of intracellular staining, cells were fixed and permeabilized using Cytofix/Cytoperm (BD). Cells were then stained with primary and, when appropriate, secondary antibodies. Propidium iodide (PI) or DAPI (4′,6-diamidino-2-phenylindole) were used as viability markers. Samples were run on a FACScalibur or Fortessa flow cytometer (Beckman Coulter) and analyzed using the FlowJo software (v7.6, TreeStar).

**Antibodies for flow cytometry**. Cells were stained using the following primary antibodies: APC-conjugated anti-mouse CD45.1 (clone A20; 553128; eBioscience), PE-conjugated or APC-conjugated anti-mouse Mac1/CD11b (clone M1/70; BD), PE-conjugated anti-mouse Ly6G/Gr1 (clone RB6-8C5; 553311/553312; BD) or PE-conjugated anti-mouse Ly6G/Gr1 (clone 1A8; 551461; BD), PE-conjugated or APC-conjugated anti-mouse CD19 (clone 1D3; 557399/561738; BD), PE-conjugated anti-mouse CD45R/B220 (clone RA3-6B2; 553089; BD), PE-conjugated anti-mouse CD3 (clone 17A2; 560527; BD) or PE-conjugated anti-mouse CD3e (clone 145-2C11; 553064; BD), PE-conjugated anti-mouse CD4 (clone GK1.5, 557308; BD), APC-conjugated anti-mouse CD8a (clone 53-6.7; 561093; BD), PE-conjugated anti-mouse CD71 (clone C2; 553267; BD), APC-conjugated anti-mouse Ter119 (clone Ter119; 116212; BioLegend), and PE-conjugated anti-mouse CD41 (clone MWReg30; 133904, BioLegend). Flow cytometric analysis of hematopoietic stem and progenitor cells was performed on a BD Fortessa flow cytometer. Primary antibodies used to mark mature (Lin⁻) cells were PerCP-Cy5.5-conjugated anti-mouse Gr1 (clone RB6-85C; 108428; BD), anti-mouse Ter119 (clone Ter119; 45-5921-82; eBioscience), anti-mouse B220 (clone RA3-6B2; 45-0452-80; eBioscience), anti-mouse CD3 (clone 172A; 560527; BD), anti-mouse CD4 (clone RM4-5; 550954; BD), anti-mouse CD8a (clone 53-6.7; 551162; BD), and anti-mouse IL-7R (clone A7R34; 45-1271-82; eBioscience). Primary antibodies used to differentiate myeloid progenitors were PECy7-conjugated anti-mouse Sca1 (clone D7; 558162; BD), APC-conjugated anti-mouse c-Kit (clone 2B8; 17-1171-83; eBioscience), APC-Cy7-conjugated anti-mouse CD16/32 (clone 93; 14-0161-81; eBioscience), FITC-conjugated anti-mouse CD34 (clone X00920; 11-0341-85; eBioscience), and biotinylated anti-mouse CD34 (clone RAM34; 13-0341-85; eBioscience) with Streptavidin-PE-TexasRed (551487; BD) or Streptavidin-E-CF594 (562318, BD). HSCs were detected using PE-conjugated anti-mouse Flk-2/Flt3 (clone A2F10.1; 553842; BD). To detect intracellular phosphorylated Smad proteins, we used PE-CF594-conjugated anti-mouse Smad2 (pS465/pS467)/Smad3 (pS423/pS425) (562697; BD).

**Viral vectors for bone marrow transduction**. To generate retroviral producer cell lines, the Phoenix™-Ampho cell line was transiently transfected with MSCV-IRES-YFP (MIY), or MSCV-IRES-GFP (MIG) or MIG-*DAB2* (isoform 1, P98082-1, verified by Sanger sequencing) using TransIT®-LT1 Transfection Reagent (Mirus Bio). Retroviral supernatants were collected, filtered through a 0.45 μm filter, and supplemented with 8 μg/ml Polybrene (Sigma), before applying to the ecotropic viral packaging cell line GP + E86. Cells were sorted by fluorescent-activated cell sorting for YFP or GFP using a BD FACSAria™ III or BD Influx II flow sorter. Marrow cells were co-cultured with stably transduced GP + E86 cells for 72 h and sorted for YFP/GFP. MND-PGK-YFP constructs that express short hairpin RNA for the knockdown of mouse *Dab2* (sh*Dab2*: GAGTGAATTGTCAGCAGACTTGG) or *Smad3* (sh*Smad3*: CTTTGACGAAGCTCATACGGA), and a negative control (shCtl) were also generated, as were MND-PGK-GFP constructs that express short hairpin RNA for knockdown of human *DAB2* (sh*DAB2*: TACAGGTTGAGAAGAAGCCAC) and a scrambled control (shCtl). Lentiviral constructs were transiently transfected using calcium phosphate into HEK293T together with one envelope plasmid (VSVG) and two packaging plasmids (ΔR and REV). Supernatants were 0.4 μm filtered and concentrated by ultracentrifugation. For the knockdown of *Dab2* or *Smad3*, sorted Lin⁻ c-Kit⁺ mouse marrow cells

were incubated with concentrated lentivirus and 8 μg/ml Polybrene for 72 h before sorting for YFP. For knockdown of *DAB2*, MDS-L cells were transduced with concentrated lentivirus and cells expressing GFP were isolated for xenotransplantation.

**Bone marrow transplantation.** To enrich for HSPCs, Pep3b donor mice were injected with 5-fluorouracil at a dose of 150 mg/kg. After 4 days, mice were euthanized by $CO_2$ inhalation and marrow cells were harvested from femurs, tibiae, and pelvic bones. Marrow cells were cultured in DMEM media containing 15% FBS, 6 ng/ml mouse interleukin-3 (m-IL-3), 10 ng/ml human interleukin-6 (h-IL-6), and 100 ng/ml murine stem cell factor overnight. Marrow cells were retrovirally transduced by co-culture with stably transduced GP + E86 cells and sorted for YFP/GFP. Alternatively, marrow was sorted for LSK cells and transduced with lentivirus, followed by sorting for YFP. Recipient C2J (Ly45.2) mice were lethally irradiated with 810 cGy by X-ray prior to being intravenously injected with $3 \times 10^5$ marrow cells or $25 \times 10^3$ LSK cells and $1 \times 10^5$ unmanipulated "helper" marrow cells from a sacrificed C2J mouse. Irradiated mice were housed with drinking water containing HCl and ciprofloxacin for 2 months. For LDAs using *miR-143/145*$^{+/+}$ or *miR-143/145*$^{-/-}$ marrow, marrow was isolated from the femurs and tibiae of untreated 12-week-old mice. Selected doses ($1 \times 10^4$, $2 \times 10^4$, $1 \times 10^5$, $5 \times 10^5$, $1 \times 10^6$) were injected into lethally irradiated C2J mice along with $1 \times 10^5$ unmanipulated helper cells. In secondary LDA transplants, marrow from the femurs and tibiae of primary mice was harvested, pooled, and transplanted into secondary recipients. The doses used (1:100, 1:500, 1:2500, 1:10,000) represent fractions of the total marrow isolated from transplanted primary mice. For competitive transplants with MIY and MIG-*DAB2*, $2.5 \times 10^5$ cells from each group were transplanted into the same recipient mouse along with $5 \times 10^4$ helper cells. In all LDAs, positive engraftment is defined as >1% repopulation in the donor GM (Mac1$^+$ and/or Gr1$^+$) population. For xenografts, female NRG-3GS mice at 10–13 weeks of age were sublethally irradiated and injected with $7.5 \times 10^5$ MDS-L cells per mouse, and engraftment was assessed by monitoring GFP$^+$ hCD33$^+$ cells in the peripheral blood.

**CFU assay.** To assess clonogenic progenitor frequencies, $3 \times 10^4$ whole marrow cells, $5 \times 10^2$ 5-fluorouracil-enriched marrow cells or $3 \times 10^2$ sorted Lin$^-$c-Kit$^+$ cells were plated in methylcellulose containing m-IL-3, h-IL-6, h-erythropoietin, and m-SCF (M3434; STEMCELL Technologies). Where indicated, cells were treated with 10 μM SMAD3 inhibitor SIS3 (Millipore) or dimethyl sulfoxide vehicle. Colonies were scored 8–10 days later. For replating assays, marrow cells were scraped off using a cell scraper and collected in a conical tube containing PBS. Cells were washed and counted before replating onto fresh methylcellulose in equal proportions per condition, and normalized to the number of input cells.

**LTC-IC assay.** To determine the frequency of LTC-IC, unsorted marrow cells from primary transplant mice were cultured for 5 weeks at limiting dilution ($3 \times 10^4$, $1 \times 10^4$, $3 \times 10^3$, $3 \times 10^3$) on a supportive feeder layer of irradiated mouse fibroblast AFT024 cells in MyeloCult (M5300; STEMCELL Technologies) media supplemented with $1 \times 10^{-6}$ M hydrocortisone. Harvested cells were plated in methylcellulose (M3434; STEMCELL Technologies) and CFU output was assessed after 12 days.

**Histology and immunohistochemistry.** Femurs were fixed in 4% paraformaldehyde (PFA), decalcified, and paraffin-embedded. Livers and spleens were fixed in 4% PFA and paraffin-embedded. Hematoxylin and eosin (H&E) staining of sections was performed. Alternatively, freshly cut tissues were analyzed for CD11b expression and immunohistochemistry was performed using the Ventana DiscoveryXT platform. In brief, tissue sections were incubated in Tris EDTA buffer at 95 °C to retrieve antigenicity, followed by incubation with the anti-CD11b primary antibody (ab133357, Abcam). Bound antibodies were incubated with secondary horse radish peroxidase-conjugated antibodies which was followed by Chromomap DAB detection. Myeloperoxidase was stained without any retrieval and detected via Optiview. Blood smears and marrow cytospins were stained with Wright–Giemsa stain.

**Real-time PCR and droplet digital PCR.** RNA was isolated using TRIzol reagent according to the manufacturer's protocol (Life Technologies). For mRNA targets, cDNA was synthesized using SuperScript II reverse transcriptase reagent and random primers (Invitrogen). Quantitative reverse transcription PCR was carried out using FastStart Universal SYBR Green Master Kit (Roche Applied Science). Alternatively, Droplet Digital PCR was performed following Bio-Rad guidelines. Results were normalized to *Gapdh* or *β-Actin* expression. For miRNA targets, cDNA was synthesized using the TaqMan miRNA Reverse Transcription Kit (Applied Biosystems). TaqMan Universal Master Mix II was used to assess miRNA expression. Real-time PCR of miRNAs was performed using TaqMan probes against 5p or 3p strands of miR-143 or miR-145 (miRBase Accession Number, MIMAT0000437, MIMAT0000157, MIMAT0017006, MIMAT0004601). miRNA expression was normalized to sno202 (mouse) or U47 (human) expression.

**Gene expression omnibus and GSEA.** Gene set enrichment analysis (GSEA)[17] was performed using published data[16] (GSE19429), using samples with del(5q) karyotype ($n = 47$) and healthy controls ($n = 17$) as the compared phenotypes.

Default parameters were used except the lower limit for gene set membership was changed to 10. Gene sets were chosen from the GO:Biological processes (BP) collection using the keyword search "SMAD* AND TGF*," resulting in 365 gene sets chosen; after filtering for gene sets with between 10 and 500 members, 239 gene sets were obtained for analysis. Leading edge analysis was performed using the top gene sets (FDR ≤ 0.05).

**Statistical analyses.** Experiments were replicated in the laboratory at least three times and the given sample numbers represent biological replicates. No blinding was done. For bone marrow transplants, experimental and control cages were randomly assigned so that each groups would have similar number of mice, with similar age and sex distribution. Animals were excluded from analyses if sick reports were attributed to known causes that were unrelated to hematopoietic issues, including fight wounds and malocclusion. Statistics were performed using GraphPad Prism (v6.0, GraphPad) and all tests were two-sided. Statistical tests were chosen based on whether our data met the assumptions, including normality and homogeneity of variances. Unpaired group comparisons were performed using Student's *t* test (Figs. 1c, d, 3a, 4a, b, 5e, 7a and 8a–f, i–j and Supplementary Fig. 1c, 4c, 4e, 5c and 6a). In Figs. 3b, 4d–f and 8e–f and Supplementary Fig. 3b, paired group comparisons were performed using Student's *t* test. Multiple *t* tests with Holm–Sidak multiple comparison were done in Fig. 4d. One-way analysis of variance (ANOVA) was used to analyze data in Figs. 1a, 3d and 5c, d, f and Supplementary Figs. 3d, e, 4c and 7. Two-way ANOVA was used to analyze Figs. 4g, 5f and Supplementary Fig. 1c. When comparing between multiple variables, correction was applied. Survival differences were assessed using the Kaplan–Meier test and log-rank test (Figs. 5b and 6a). LDAs (Figs. 1e, 4h and 5g) were analyzed using extreme limiting dilution analysis[42]. The relationship between two variables was assessed by Pearson's correlation and linear regression (Fig. 7d–g and Supplementary Figs. 1a, b and 4f).

**Data availability.** All relevant datasets are available online from referenced source.

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

## Acknowledgements

This work was supported by grants to A.K. from the Terry Fox Research Institute, the Canadian Institutes of Health Research (CIHR), the Leukemia and Lymphoma Society of Canada, and the Cancer Research Society. The following agencies provided salary support: CIHR (J.L., J.W., L.C., R.I.), European Molecular Biology Organization (J.W.), US Department of Defense (L.C.), the Michael Smith Foundation for Health Research (J.W., L.C.), the University of British Columbia (J.L.), Natural Sciences and Engineering Research Council (K.S.), and the Centre for Blood Research (K.S.), A.K. is the recipient of the John Auston BC Cancer Foundation award.

## Author contributions

J.L. and M.v.d.B. performed experiments, analyzed data, and wrote the paper. J.W. performed experiments and analyzed data. J.P. analyzed data and wrote the paper, R.I., K. S., L.C., S.M.-H., Y.D., P.U. and M.F. performed experiments. G.C. and M.B. provided gene-targeted mice. A.K. conceived the project, analyzed data, and wrote the paper.

## Additional information

**Competing interests:** The authors declare no competing interests.

