## [Peer Review File · Nature Communications]

Reviewers' comments:

Reviewer #1 (Remarks to the Author):

The role of microRNAs in the regulation in hematopoietic cell production remains poorly understood. Although there are now multiple mir's that have shown to have regulatory activity in the hematopoietic system (155, 125), this picture is far from complete. Therefore, the findings of the authors are potentially of great interest. As it would have been surprising that enforced expression of a single mir-145 target, DAB2, would recapitulate the phenotype seen in mir145 $-/-$ mice, I was anxious to read the paper. Unfortunately, I was not impressed by the data, and in fact believe that there is no, or only a very minor role for mir143/145 in regulating hematopoiesis.

The authors write in their Discussion that "The current study provides direct evidence that miR-143 and miR-145 are able to affect the hematopoietic system at the stem cell level through the TGF β pathway and points to a previously unappreciated role for DAB2 in HSPC function". I believe that this conclusion exaggerates markedly the findings presented in the paper. In all fairness, the results that the authors report are very modest, and often do not reach statistical significance.

The authors write that 143/145 $-/-$ mice live shorter, but in fact the data presented in Figure 1 do not substantiate this. The data clearly show that WT and 143/145 $-/-$ live equally long.

The statistics in Figure 2a for the LT-HSCs are odd. Heterozygous 143/145 $+/-$ mice are not significantly different compared to wt, whereas 143-145 $-/-$ are, although on average LT-HSC numbers are lower in the hets. Also the spread of values appears to be higher in the hets. As the authors interpret these data as biologically relevant, it requires a very solid analysis. I am not convinced that there are major differences in LT-HSCs numbers, which is also substantiated by Figure 2e. In the text the authors claim that the data in Figure 2e show significant differences ($p < 0,026$), but the figure clearly shows overlapping 95% confidence intervals. This to me does not suggest significant differences.

The identification of Dab2 was based on expression in patient samples. I do not understand why the authors did not verify that this gene, and others, were upregulated in LT-HSCs isolated from 143/145 $-/-$ mice. The only attempt to this end is provided in Figure 4C, where data are not strong. There are no replicates, the quantification is tricky, and there is no information on sensitivity/specificity of the antibody. They have used a very clean model system which allows them to precisely ask which genes are differentially expressed (or even which proteins are differentially abundant), but they chose not to use this model and instead use much more heterogeneous patient samples. This does not make sense to me. If Dab2 is really an important downstream effector of 143/145 in murine LT-HSCs, its expression will be affected, and this should be measured.

The data in Figure 5a convincingly show that overexpression of DAB2 reduces long-term reconstitution potential, at least when competing with wt cells. In the primary recipients the effect on different cell subsets in BM is quite variable, and therefore the data are borderline -or not- significant. This also applies to secondary transplants.

In the text describing the LDA data of Figure 5h the authors write that there is a significant difference ($p < 0,036$), but again, the 95% CI limits overlap. In general, all data presented in figure 5e and f are not, or borderline significant. This is not different from the data in which 143/154 $-/-$ mice were used, so in that sense the Dab2 data indeed are similar to those in 143/145 mice, but it is all not very convincing.

To correctly interpret the data in Figure 6 it needs to be clear how exactly the experiment was

executed. Particularly, in how many secondary mice were primary DAB2 (pre) malignant cells transplanted, and reversely, from how many primary mice were the BM cells derived that were used in the 8 secondary mice?

Were all Dab2-CE mice in Figure 6b transplanted from a single pool of transduced cells, or are these data the result of multiple independent experiments?

Although Dab2-CE transduced BM cells show an 8-fold increase in LT-HSC activity in LDA analysis, the 95% CI limits between the two groups are -again- overlapping.

I do not believe that old 143/154 $-/-$ mice develop thrombocytopenia. The data show that plt counts (for unclear reasons) go up in old wt mice, and are stable in old 143/145 mice. I'm sure there is no significant thrombocytopenia as 143/145 mice age, and the statistical comparison is probably done comparing old wt with old 143/145 mice. Similarly, the authors claim in the text that aged 143/145 $-/-$ mice become anemic, but in fact Figure 7a does not show any significant differences between the groups at the latest time point.

Reviewer #2 (Remarks to the Author):

The manuscript reports a follow up work of the same lab on the role of microRNAs located on chromosome 5q in the pathogenesis of myelodysplastic syndromes. In particular, they focus on miR-143/145, both co-expressed as a single primary transcript, and located within the commonly deleted region in del(5q) MDS resulting in haploinsufficiency. MiRNA loss at the 5q locus is restricted to miR-143/145 in about 60% of MDS patients with del(5q), while 40% have a larger deletion including also miR-146a, possibly a feature of more aggressive disease, and the subject of previous papers. Using a miR-143/miR-145 knockout mouse model, the authors show that mice have a reduced LT-HSC content and a slightly shorter long-term survival, probably due to the development of a low-penetrance myeloproliferative syndrome late in life. Homozygous 143/145 knockout mice show significantly higher white blood cell counts as well as lower Hgb and Plt levels compared to controls, reflecting single mice in the cohort that developed hepatosplenomegaly and leukocytosis. Mechanistically, the authors associate this phenotype to hyperactivation of the TGFbeta pathway that appears to be targeted at multiple levels by miR-143/145. In particular, they validate DAB2, an adapter protein central to TGFbeta signaling, as a direct miR-145 target and perform an extensive set of experiments that consolidate a role of chronically activated TGFbeta signaling on hematopoiesis predisposing to myeloid malignancy. While TGFbeta signaling has been previously associated to HSC quiescence (in line with the finding of reduced HSC numbers and reduced repopulating potential of miR-143/145 $-/-$ cells or cells constitutively expressing DAB2), this work highlights a differential effect of chronically activated TGFbeta signaling on myeloid progenitor cells leading to their expansion in some but not all mice. The work is technically solid, well written and of interest. However, some points should be further addressed.

1. The mechanism behind the differential impact of TGFbeta signaling on myeloid progenitors (as opposed to HSC) remains unclear. The long latency and incomplete penetrance of myeloproliferation suggest that cooperating events/indirect effects (rather than TGFbeta activation in GMP as the primary event) contribute to this phenotype. Can the authors clarify over what timeframe the myeloid expansions develop, e.g. by looking at the serial bleedings that were performed in the DAB2-CE mice? Does myeloproliferation correlate with the level of ectopic DAB2 expression in the single mice? It is unclear why DAB2 levels correlate inversely with miR-145 expression in BM subpopulations from mice transplanted with DAB2-CE cells (Suppl. Fig 6c,d). Does the Dab2 cDNA used in the vector contain the

3'UTR sequence, or do the authors measure endogenous Dab2 rather than vector-derived Dab2? These studies should be repeated in WT mice and 143/145 knockout mice. Why is the incidence and severity of myeloproliferation much lower in the miR-143/145 knockout mice compared to the DAB2-CE mice? Some dose correlation between the level of TGFbeta pathway activation and the degree of myeloproliferation should be provided, to potentially strengthen a direct mechanistic link between this pathway and oncogenesis.

2. Did some of the miR-143/145 knockout mice develop a transplantable myeloid neoplasia, similar in aggressiveness to the DAB2 CE mice? Given that constitutive DAB2 expression has been obtained by transduction with a potentially oncogenic retroviral vector, one wonders whether insertional mutagenesis may have contributed to the malignant phenotype in the latter setting. What was the transduction efficiency with the DAB2 CE and YFP control vector, respectively, in vitro and in the mice with MPD?

3. The authors hypothesize that the GMP population represents the cell of origin for myeloproliferative disease. Another interpretation could be that functional alterations in HSC (triggered by hyperactive TGFbeta signaling in this compartment rather than in GMPs) lead to the accumulation of cooperating mutations which, at a certain point in time, confer self renewal potential to a downstream GMP population which progressively expands and gives rise to MPD. Some more discussion, ideally supported by additional data, is warranted.

4. A link between DAB2 and miR-145 has been well demonstrated, including in vitro rescue experiments in colony-forming cells. Given the centrality of the TGFbeta pathway in this work, additional experimental data consolidating the link between miR-143/145 and this pathway would strengthen the manuscript. Could any of the additional predicted targets for miR-143 and/or miR-145 within the TGFbeta pathway be validated? The combinatorial targeting of multiple targets linked in a signal transduction cascade could help explain why/how haploinsufficiency at the 5q locus could lead to such far-reaching biological consequences.

5. The authors show that DAB2 knockdown in a human MDS 5q- cell line shifts the CD34+ compartment to a higher proportion of CD38- cells. Were overall engraftment levels of the MDS cells influenced by DAB2 knockdown, and what was the functional consequence of restricting TGFbeta activation? What is the impact of constitutive DAB2 expression on normal human CD34+ hematopoietic stem and progenitor cells?

6. Some of the claims made in the text are not appropriate and should be toned down or revised. On page 6, the authors hypothesize that "HSC activity also appeared to be reduced in miR-143/145+/- mice..". This hypothesis is based on CFU assay, which does not provide an HSC readout-in addition to a p value of 0.43 which is far from significant. On page 8, they claim that DAB2 did not affect progenitor activity in primary CFU assays, while Fig. 5a shows a statistically significant reduction. On page 9 (line 9), the relative increase in GMPs within DAB2-GFP cells with respect to the YFP control cell compartment is not fully convincing, given that all progenitor compartments show a similar trend (with variability between mice), rather suggesting that the DAB2-GFP compartment shows a reduction in Lineage+ cells. It may be problematic to compare subpopulations within the rare GFP+ cell compartment (comprising on average <10% at 20 weeks) to the much bigger YFP compartment. On page 11, the authors say that "miR-143/145-/- mice showed leukocytosis, thrombocytopenia and anemia", while the group averages referred to in Fig.7a show WBC, HGB and PLT levels within the normal range, even in the the 143/145-/- group.

Reviewer #3 (Remarks to the Author):

Karson et al show that DAB2 is a target of micrnas deleted in 5q- MDS. They show that increased expression of DAB2 due to 5q deletion leads to activation of TGFb-smad signaling and leads to stem and progenitor defects in vivo.

The studies are comprehensive and elegantly done. The use of multiple mouse models support the conclusions. The findings provide a link between a genetic deletion and activation of TGF signaling in MDS and advance the field.

Some minor concerns:

1. The gapdh loading control in the western blots appears to be from a different membrane.
2. papers showing activation of smad2/3 in human mds can be cited for completion
3. List of all TGF pathway genes that were altered and are micrrna targets should be moved to main figures instead of supplementary

Reviewer #1 (Remarks to the Author):

'The role of microRNAs in the regulation in hematopoietic cell production remains poorly understood. Although there are now multiple mir's that have shown to have regulatory activity in the hematopoietic system (155, 125), this picture is far from complete. Therefore, the findings of the authors are potentially of great interest. As it would have been surprising that enforced expression of a single mir-145 target, DAB2, would recapitulate the phenotype seen in mir145 -/- mice, I was anxious to read the paper. Unfortunately, I was not impressed by the data, and in fact believe that there is no, or only a very minor role for mir143/145 in regulating hematopoiesis. The authors write in their Discussion that "The current study provides direct evidence that miR-143 and miR-145 are able to affect the hematopoietic system at the stem cell level through the TGFβ pathway and points to a previously unappreciated role for DAB2 in HSPC function". I believe that this conclusion exaggerates markedly the findings presented in the paper. In all fairness, the results that the authors report are very modest, and often do not reach statistical significance.'

As the reviewer notes, the effect size of the phenotype we see is small, but statistically significant. The small effect recapitulates the disease phenotype in del(5q) myelodysplastic syndrome very well. In the human disorder, deletion of chromosome 5q and the accompanying depletion of miR-143 and miR-145 results in very good outcome disease with the highest median survival of any myelodysplastic syndrome. Thus our findings of the halving of the stem cell frequency with a late myeloproliferative effect is concordant with human disease, and we consider these findings to provide novel information on the role of miR-143/145 in normal and malignant hematopoiesis.

'The authors write that 143/145-/- mice live shorter, but in fact the data presented in Figure 1 do not substantiate this. The data clearly show that WT and 143/145-/- live equally long.'

We agree with Reviewer #1 that although we followed a large number of mice (WT n = 20, KO n = 29), the modest effect size of miR-143/145 deficiency means that the survival data may not be sufficiently powered. Given that wild-type mice are also approaching the end of their lifespan the number of animals and time required to power this question, we feel that this is beyond the scope of the current manuscript. However, what we noted but did not explicitly point out in the initial manuscript is that 3 out of 6 miR-143/145-deficient mice died of a myeloproliferative disorder while none of the moribund wild-type animals showed a myeloproliferative disorder ($P = 0.05$, Fisher's exact test). This finding has been added to the Results section (p. 12, line 1).

‘The statistics in Figure 2a for the LT-HSCs are odd. Heterozygous 143/145+/- mice are not significantly different compared to wt, whereas 143-145 -/- are, although on average LT-HSC numbers are lower in the hets. Also the spread of values appears to be higher in the hets. As the authors interpret these data as biologically relevant, it requires a very solid analysis. I am not convinced that there are major differences in LT-HSCs numbers, which is also substantiated by Figure 2e. In the text the authors claim that the data in Figure 2e show significant differences (p<0,026), but the figure clearly shows overlapping 95% confidence intervals. This to me does not suggest significant differences.’

We acknowledge the higher spread of values in the heterozygote mice and thank Reviewer #1 for pointing out an error in the legend, which we have now corrected. The box-and-whisker plots in Figure 2a from the manuscript (Fig. 1a in the revised manuscript) show the medians, rather than the averages. When representing these data as a bar graph (Response Fig. 1), it shows that the mean % LT-HSC is slightly higher in the heterozygote group. Nevertheless, it should be pointed out that the *P* value of 0.07 indicate that for heterozygous mice the likelihood that these differences are sufficient to reject the null hypothesis is 0.07 versus 0.04 for the miR-143/145^{-/-} mice (*P* = 0.04). Thus we agree with the reviewer that the effect is likely similar for miR-143/145^{+/-} mice, and have indicated this in the text as a trend towards lower LT-HSC for the heterozygous mice (p. 5, line 25).

Response Figure 1. Loss of miR-143/145 results in reduced LT-HSC frequency. Frequency of long-term-HSC (LT-HSC, CD45⁺EPCR⁺CD48⁻CD150⁺) in the marrow of 8-12 week old wild-type (WT), miR-143/145^{+/-} and miR-143/145^{-/-} mice, as analyzed by flow cytometry (mean ± SEM, WT n = 11, miR-143/145^{+/-} n = 9, 143/145^{-/-} n = 8). Data is shown as mean ± SEM).

With respect to overlapping confidence intervals in Fig. 2e (Fig. 1e in the revised manuscript), it is a common misconception that if 95% CI overlap, the difference cannot be significant at the 5% level. In fact the *P* value takes into account the variability of the data, and many highly impactful studies show similar overlaps in 95% CI. We refer Reviewer #1 to the short overview published on this topic by Wolfe & Hanley (CMAJ, 2002; 166(1)).

‘The identification of Dab2 was based on expression in patient samples. I do not understand why the authors did not verify that this gene, and others, were upregulated in LT-HSCs isolated from 143/145-/- mice. The only attempt to this end is provided in Figure

4C, where data are not strong. There are no replicates, the quantification is tricky, and there is no information on sensitivity/specificity of the antibody. They have used a very clean model system which allows them to precisely ask which genes are differentially expressed (or even which proteins are differentially abundant), but they chose not to use this model and instead use much more heterogeneous patient samples. This does not make sense to me. If Dab2 is really an important downstream effector of 143/145 in murine LT-HSCs, its expression will be affected, and this should be measured.'

We agree with the reviewer and have performed additional western blotting to verify that Dab2, and other miR-143/145 targets in the TGF β pathway, are upregulated in 143/145^{-/-} mouse marrow. We repeated the western blots for Dab2 using wild-type and miR-143/154^{-/-} lineage-negative marrow in replicate (new Fig. 3c in the revised manuscript). In addition, we analyzed protein expression of Tgfr2, Rbpms, Sara, Smad2-5, Tak1, Map2k4 and p38 (new Supplementary Fig. 2 in the revised manuscript). Data from at least three independent experiments were quantified by densitometry and *P* values are indicated where significant. Dab2 protein was significantly upregulated (1.8 fold compared to wild-type, *P* = 0.001). The upstream signaling molecules Rbpms (*P* = 0.003) and Sara (*P* = 0.05) were also significantly increased in 143/145^{-/-} lineage-negative marrow. These data are described in the Results section on p. 7 from line 23.

'The data in Figure 5a convincingly show that overexpression of DAB2 reduces long-term reconstitution potential, at least when competing with wt cells. In the primary recipients the effect on different cell subsets in BM is quite variable, and therefore the data are borderline -or not- significant. This also applies to secondary transplants. In the text describing the LDA data of Figure 5h the authors write that there is a significant difference (p<0,036), but again, the 95% CI limits overlap. In general, all data presented in figure 5e and f are not, or borderline significant. This is not different from the data in which 143/154^{-/-} mice were used, so in that sense the Dab2 data indeed are similar to those in 143/145 mice, but it is all not very convincing.'

We agree with the reviewer that there is variability in the data of Fig. 5f (Figure 4f in the revised manuscript), and this correlates very well with the fact that only a proportion of *DAB2*-expressing mouse chimeras go on to a myeloproliferative disorder, and is quite consistent with the thesis of our paper. In fact the proportion of cases where *DAB2*-GFP cells show increased chimerism is the same 20 - 25% as the proportion of animals that go on to a myeloid malignancy. If the two animals that are considered to be outliers (CMP (FDR 2%), GMP (FDR 10%) and MEP (FDR 10%), ROUT test) are removed from the analysis, it is clear that *DAB2* cells are less competitive in this *in vivo* assay (Response Fig. 2 below), except for those that go on to a myeloid malignancy.

Response Figure 2. DAB2 affects reconstitution potential. Bone marrow was harvested at 20 weeks to assess GFP/YFP chimerism (mean, $n = 6$, 2 outliers removed) of LSK ($\text{Lin}^{-}\text{Sca1}^{+}\text{c-Kit}^{+}$) and myeloid progenitors (CMP, $\text{Lin}^{-}\text{Sca1}^{\text{c-Kit}^{+}}\text{CD34}^{+}\text{CD16/32}^{\text{lo}}$; GMP, $\text{Lin}^{-}\text{Sca1}^{\text{c-Kit}^{+}}\text{CD34}^{+}\text{CD16/32}^{\text{hi}}$; MEP, $\text{Lin}^{-}\text{Sca1}^{\text{c-Kit}^{+}}\text{CD34}^{+}\text{CD16/32}^{\text{lo}}$).

Reviewer #1 also had concerns regarding the small effect sizes seen in this model and the associated statistics in Fig. 5h from the manuscript (4h in the revised manuscript). In fact, the point of this manuscript is that many of the findings seen in myelodysplastic syndrome with del(5q), which are haploinsufficient for miR-143/145, are subtle and consistent with what we are seeing in this mouse model. With respect to overlapping confidence intervals it should be noted that the P values take into account the spread of the data. The point of the probability function is to determine the likelihood that the null hypothesis is rejected. We have chosen, as many investigators do, a probability of 5% of the null hypothesis being correct as sufficient evidence to reject the null hypothesis. It is a common misconception that if 95% CI overlap, the difference cannot be significant at the 5% level and we refer Reviewer #1 to the short overview published on this topic by Wolfe & Hanley (CMAJ, 2002; 166(1)). Again, the small effect size we see in the mouse model is entirely consistent with the counterpart in human disease.

‘To correctly interpret the data in Figure 6 it needs to be clear how exactly the experiment was executed. Particularly, in how many secondary mice were primary DAB2 (pre) malignant cells transplanted, and reversely, from how many primary mice were the BM cells derived that were used in the 8 secondary mice? Were all Dab2-CE mice in Figure 6b transplanted from a single pool of transduced cells, or are these data the result of multiple independent experiments?’

With respect to Figure 6 from the manuscript, we apologize that the experimental details were not clear. We have expanded on this in our revised manuscript (p. 9, line 23 and p. 10, line 20). In brief, three independent primary transplants ($n = 6$, $n = 4$, $n = 8$) were performed using a pool of transduced cells. Two independent secondary transplants (each $n = 4$) were performed using unsorted cells from moribund DAB2-AML/MPD mice.

‘Although Dab2-CE transduced BM cells show an 8-fold increase in LT-HSC activity in LDA analysis, the 95% CI limits between the two groups are -again- overlapping.’

With respect to overlapping confidence intervals in Fig. 6g (Fig. 5g in the revised manuscript) , it should be noted that it is a common misconception that if 95% CI overlap, the difference cannot be significant at the 5% level and we refer Reviewer #1 to the short overview published on this topic by Wolfe & Hanley (CMAJ, 2002; 166(1)).

'I do not believe that old 143/154 $-/-$ mice develop thrombocytopenia. The data show that plt counts (for unclear reasons) go up in old wt mice, and are stable in old 143/145 mice. I'm sure there is no significant thrombocytopenia as 143/145 mice age, and the statistical comparison is probably done comparing old wt with old 143/145 mice. Similarly, the authors claim in the text that aged 143/145 $-/-$ mice become anemic, but in fact Figure 7a does not show any significant differences between the groups at the latest time point.'

Since our original submission, we have analyzed data from more mice. Blood counts from a larger cohort of mice aged 80 weeks (WT n = 28, KO n = 23) showed a significant relative increase in leukocytes ($P = 0.03$) and decreased hemoglobin ($P = 0.05$) and platelets ($P = 0.05$) (new Fig. 7a in the revised manuscript). As the reviewer points out, miR-143/145 $^{-/-}$ group means are still within the normal range (Mazzaccara et al. PLoS 2008). Nevertheless, a larger proportion of miR-143/145 $^{-/-}$ mice displayed leukocytosis (WBC > $14 \times 10^3/\text{mm}^3$, 17% in miR-143/145 $^{-/-}$ vs 7% in WT). These data are described in the Results section on p. 11, line 28, and are consistent with the DAB2 experiments showing that only a proportion of mice develop myeloid malignancy.

Reviewer #2 (Remarks to the Author):

'The manuscript reports a follow up work of the same lab on the role of microRNAs located on chromosome 5q in the pathogenesis of myelodysplastic syndromes. In particular, they focus on miR-143/145, both co-expressed as a single primary transcript, and located within the commonly deleted region in del(5q) MDS resulting in haploinsufficiency. MiRNA loss at the 5q locus is restricted to miR-143/145 in about 60% of MDS patients with del(5q), while 40% have a larger deletion including also miR-146a, possibly a feature of more aggressive disease, and the subject of previous papers. Using a miR-143/miR-145 knockout mouse model, the authors show that mice have a reduced LT-HSC content and a slightly shorter long-term survival, probably due to the development of a low-penetrance myeloproliferative syndrome late in life. Homozygous 143/145 knockout mice show significantly higher white blood cell counts as well as lower Hgb and Plt levels compared to controls, reflecting single mice in the cohort that developed hepatosplenomegaly and leukocytosis. Mechanistically, the authors associate this phenotype to hyperactivation of the TGFbeta pathway that appears to be targeted at multiple levels by miR-143/145. In particular, they validate DAB2, an adapter protein central to TGFbeta signaling, as a direct miR-145 target and perform an extensive set of experiments that consolidate a role of chronically activated TGFbeta signaling on hematopoiesis predisposing to myeloid malignancy. While TGFbeta signaling has been previously associated to HSC quiescence (in line with the finding of reduced HSC numbers and reduced repopulating potential of miR-143/145^{-/-} cells or cells constitutively expressing DAB2), this work highlights a differential effect of chronically activated TGFbeta signaling on myeloid progenitor cells leading to their expansion in some but not all mice. The work is technically solid, well written and of interest. However, some points should be further addressed.'

'1. The mechanism behind the differential impact of TGFbeta signaling on myeloid progenitors (as opposed to HSC) remains unclear. The long latency and incomplete penetrance of myeloproliferation suggest that cooperating events/indirect effects (rather than TGFbeta activation in GMP as the primary event) contribute to this phenotype. Can the authors clarify over what timeframe the myeloid expansions develop, e.g. by looking at the serial bleedings that were performed in the DAB2-CE mice?'

We acknowledge the reviewer's point. Given the late development of malignancy in only a subset of mice, whether *DAB2-CE* or miR-143/145-deficient, it is probable that TGFβ signaling alone is not sufficient for disease development. We agree that the myeloid malignancy we observed in *DAB2-CE* mice may be the result of insertional mutagenesis or additional stochastic mutations that accumulate over time. This could, in addition to the higher level of canonical TGFβ pathway activation, explain the increased incidence and severity of myeloproliferation in the *DAB2-CE* mice than in the miR-143/145^{-/-} mice. It would be of interest in future studies to identify collaborating events that contribute to disease development in these mice, but we believe it is beyond the scope of our current manuscript.

We have also now presented the timeframe over which the hematologic parameters change using serial hematology data we collected from *DAB2*-CE mice. From 28 weeks after primary transplant, we observed leukocytosis, anemia and thrombocytopenia (new Supplementary Fig. 4d in the revised manuscript) in 7 out of 19 *DAB2*-CE mice that developed a myeloid malignancy, and the concordance of the blood counts in specific mice are displayed visually. These data are described in the Results section on p. 9 from line 28.

‘Does myeloproliferation correlate with the level of ectopic *DAB2* expression in the single mice?’

We analyzed the mean fluorescence intensity of GFP⁺ *DAB2*-CE cells in all *DAB2*-CE mice in the peripheral blood as a surrogate for the level of ectopic *DAB2* expression at 24 weeks post-transplant. Comparing the geometric mean fluorescence intensity (gMFI) between mice that did and did not develop disease, we note a trend towards increased fluorescence in the *DAB2*-AML/MPD group ($P = 0.06$) (new Supplementary Fig. 4e in the revised manuscript), although this was mainly skewed by two outliers. In addition we analyzed endpoint leukocyte counts and GFP⁺ gMFI in the individual *DAB2*-CE mice, and did not find a significant correlation ($r = 0.32$, $P = 0.09$) (new Supplementary Fig. 4f in the revised manuscript). Further, the time to transformation of animals that developed disease also did not correlate with gMFI ($r = 0.14$, $P = 0.74$). We have described these data in the revised paper (p. 10, line 5). These results suggest that ectopic constitutive expression of *DAB2*, at the level driven by the viral promoter, does not strongly correlate with development of myeloproliferative disease. Rather we posit that some threshold level of constitutive pathway activation, that is surpassed in every case in our system, is sufficient to drive the observed phenotype, and that secondary events lead to the development of malignancy in a subset of mice.

‘It is unclear why *DAB2* levels correlate inversely with miR-145 expression in BM subpopulations from mice transplanted with *DAB2*-CE cells (Suppl. Fig 6c,d). Does the *Dab2* cDNA used in the vector contain the 3’UTR sequence, or do the authors measure endogenous *Dab2* rather than vector-derived *Dab2*? These studies should be repeated in WT mice and 143/145 knockout mice.’

We apologize for the confusion and would like to clarify that in Supplementary Fig. 6 from the original manuscript endogenous *Dab2* expression was measured in BM populations from wild-type mice (not mice transplanted with *DAB2*-CE). We have emphasized this in the legend. In wild-type mice, we found that miR-145 expression was anti-correlated with *Dab2* expression (Supplementary Fig. 7a and b in the revised manuscript). We have now also measured *Dab2* mRNA expression in miR-143/145^{-/-} marrow where miR-145 is not expressed. Our new data indicate a relative increase in endogenous *Dab2* expression in the GMP and MEP populations of miR-143/145^{-/-} marrow (new Supplementary Fig. 7c in the revised manuscript).

‘Why is the incidence and severity of myeloproliferation much lower in the miR-143/145 knockout mice compared to the DAB2-CE mice? Some dose correlation between the level of TGFbeta pathway activation and the degree of myeloproliferation should be provided, to potentially strengthen a direct mechanistic link between this pathway and oncogenesis.’

As noted above, we believe that there is some threshold level of TGFβ activation that drives the malignant phenotype, which is surpassed in the DAB2-CE mice, but perhaps not in all miR-143/145^{-/-} mice. We found that the protein level of Dab2 in miR-143/145^{-/-} marrow (new Fig. 3c in the revised manuscript) is 1.8-fold increased over WT, whereas in *DAB2-CE* cells (Supplementary Fig. 4a in the revised manuscript) Dab2 protein is 2.5-fold increased compared to the Vector control. Furthermore, enforced *DAB2* expression increased phosphorylation of SMAD2/3 under basal conditions (Supplementary Fig. 3d in the revised manuscript), whereas TGFβ stimulation was necessary to observe a significant differential increase in phosphorylation of Smad2/3 in miR143/145^{-/-} marrow (Fig. 3b in the revised manuscript). Together, these data indicate a higher level of TGFβ pathway activation in *DAB2-CE* mice compared to miR143/145^{-/-}, which may be the differential threshold that explains the differential rates and incidence of transformation.

To strengthen the link between the level of TGFβ pathway activation and HSPC function we performed colony forming unit (CFU) assays using the TGFβR1 Kinase Inhibitor galunisertib (LY2157299). We used various doses of galunisertib in order to analyze the effect of various degrees of TGFβ pathway inhibition, as measured by phosphorylation of Smad2/3 (new Supplementary Fig. 9i in the revised manuscript). We observed that greater TGFβ pathway inhibition resulted in more functional recovery of miR143/145^{-/-} HSPC activity, particularly upon serial replating implying a dose-dependent effect of TGFβ pathway activation on the more primitive cells (new Fig. 8e-f in the revised manuscript). These data are described in the Results section on p. 13 from line 4.

In addition, we transplanted miR-143/145^{-/-} cells transduced with control construct (shCtl) or sh*Dab2*, followed by *ex vivo* analysis of primitive hematopoietic progenitor cells at limiting dilution. Eight weeks post-transplant, cells from the mice were harvested to start a long-term culture-initiating cell (LTC-IC) assay. HSC frequency was increased in miR-143/145^{-/-} cells with *Dab2* knockdown by approximately two-fold (new Fig. 8g and new Supplementary Table 7 in the revised manuscript), which is similar to the reduction in stem cell activity in miR-143/145^{-/-} marrow shown in Fig. 1e of the revised manuscript, suggesting that HSC dysfunction in miR-143/145^{-/-} mice can be explained by derepression of *Dab2*. These data are described in the Results section on p. 13 from line 11.

To summarize, we see a phenotypic response that is TGFβ activation level-dependent in miR-143/145^{-/-} cells, and we also see a higher level of *DAB2* expression in the *DAB2-CE* experiments than noted in miR-143/145^{-/-} cells. Taken together our findings suggest that in miR-143/145^{-/-} mice a lower level of TGFβ activation may delay development of myeloproliferation. Further, the threshold for cooperation with other mutations to drive myeloid malignancy may not

always be reached. Alternatively, or in addition, insertional mutagenesis in the *DAB2*-CE model may result in more cooperating mutations to drive myeloid malignancy in the *DAB2*-CE model.

‘2. Did some of the miR-143/145 knockout mice develop a transplantable myeloid neoplasia, similar in aggressiveness to the DAB2 CE mice?’

We observed that 3 out of 6 miR-143/145-deficient mice died of a myeloproliferative disorder with spleen and liver infiltration (Fig 7b, c in the revised manuscript) while none of the moribund wild-type animals showed a myeloproliferative disorder ($P = 0.05$, Fisher’s exact test). Because these mice died at times when we were not able to harvest viable bone marrow, we were unfortunately not able to perform secondary transplants. Given the timeframe of disease development we unfortunately were not able to examine this further with a new cohort of miR-143/145^{-/-} mice. However, the leukocytosis and massive infiltration of the spleen and livers of these mice is entirely consistent with a myeloproliferative disorder.

‘Given that constitutive DAB2 expression has been obtained by transduction with a potentially oncogenic retroviral vector, one wonders whether insertional mutagenesis may have contributed to the malignant phenotype in the latter setting.’

We acknowledge the reviewer’s point. Given the late development of malignancy in only a subset of mice, whether *DAB2*-CE or miR-143/145-deficient, it is probable that TGFβ signaling alone shows longer latency to develop disease. Thus as the reviewer suggests, the higher likelihood of developing myeloid malignancy in *DAB2*-CE mice may be the result of insertional mutagenesis or additional stochastic mutations that accumulate over time. This could, in addition to the higher level of canonical TGFβ pathway activation, explain the increased incidence and severity of myeloproliferation in the *DAB2*-CE mice than in the miR-143/145^{-/-} mice. It would be of interest in future studies to identify collaborating events that contribute to disease development in these mice, but we believe it is beyond the scope of our current manuscript.

‘What was the transduction efficiency with the DAB2 CE and YFP control vector, respectively, in vitro and in the mice with MPD?’

The transduction efficiency of mouse bone marrow cells *in vitro* was 35% for Vector and 20% for *DAB2*-CE. *In vivo*, average long-term engraftment (>16-weeks post-transplant) was 65% for Vector and 42% for *DAB2* (Supplementary Fig. 4b in the revised manuscript). *DAB2*-CE mice that developed AML/MPD had similar long-term engraftment levels to those of *DAB2*-CE mice that did not (average 47%, $P = 0.5$) (Response Fig. 3), again suggesting the requirement of secondary cooperating events that lead to malignancy. This information is described in the Results section on p. 9 from line 24.

Response Figure 3 Engraftment analysis in *DAB2-CE* mice. Percentage of GFP in peripheral blood was measured at 24 weeks post-transplant. Data from individual mice is plotted against the corresponding white blood cell count (WBC) at endpoint (mean \pm SEM).

‘3. The authors hypothesize that the GMP population represents the cell of origin for myeloproliferative disease. Another interpretation could be that functional alterations in HSC (triggered by hyperactive TGFbeta signaling in this compartment rather than in GMPs) lead to the accumulation of cooperating mutations which, at a certain point in time, confer self renewal potential to a downstream GMP population which progressively expands and gives rise to MPD. Some more discussion, ideally supported by additional data, is warranted.’

We agree with the reviewer that there are alternative possibilities regarding the cell of origin for AML/MPD in our mice. We have thus softened the statements (p. 16, line 20) regarding the population of origin for myeloproliferative disease, as we agree that our results are not sufficient to attribute the expansion of *DAB2-CE* cells to a specific stem/progenitor compartment. However, to rigorously determine the answer to this question, we feel we would have to perform secondary transplants of sorted progenitor and HSC populations of 1°*DAB2-AML/MPD* mice into secondary recipients. Given the unpredictability of primary malignancy developing and the time required to perform primary and then secondary transplants we feel that further experimentation is out-of-scope for this manuscript.

‘4. A link between *DAB2* and miR-145 has been well demonstrated, including in vitro rescue experiments in colony-forming cells. Given the centrality of the TGFbeta pathway in this work, additional experimental data consolidating the link between miR-143/145 and this pathway would strengthen the manuscript. Could any of the additional predicted targets for miR-143 and/or miR-145 within the TGFbeta pathway be validated? The combinatorial targeting of multiple targets linked in a signal transduction cascade could help explain why/how haploinsufficiency at the 5q locus could lead to such far-reaching biological consequences.’

We agree that our manuscript would benefit from validating other predicted miR-143/145 targets within the TGFβ pathway. Firstly, in response to a similar comment by Reviewer #1

(Point 4), we have performed western blotting to analyze protein expression of TGF β pathway components in progenitor-enriched wild-type and miR-143/145^{-/-} marrow. The data show that besides Dab2, Rbpms and Sara protein levels are also significantly upregulated in miR-143/145^{-/-} marrow (new Supplementary Fig. 2 in the revised manuscript). Although we did not see increased protein expression of Smad proteins, we found enhanced phosphorylation of Smad2/3 via intracellular phospho-flow cytometry (Fig. 3b in the revised manuscript), in line with enhanced TGF β pathway activation.

In contrast, we did not detect alterations in proteins of the non-canonical TGF β pathway. We also tested functionally whether the non-canonical TGF β pathway could be involved in the miR-143/145^{-/-} phenotype by performing CFU assays using the TAK1 inhibitor 5Z-7-Oxozeaenol and the p38 inhibitor SB203580. Inhibition of TAK1 did not affect primary or secondary colony formation in either WT or miR-143/145^{-/-} marrow (new Supplementary Fig. 9e-f in the revised manuscript). Inhibition of p38 resulted in an increase in colony formation upon replating, however, no differential effect was noted between WT and miR-143/145^{-/-} marrow cells when treated with this inhibitor (new Supplementary Fig. 9g-h in the revised manuscript). Therefore, we conclude that the SMAD pathway, but not the non-canonical TGF β pathway, is involved in the effect of miR-143/145 on HSPC. These findings have now been added to the Results section (p. 12, line 28).

‘5. The authors show that DAB2 knockdown in a human MDS 5q- cell line shifts the CD34+ compartment to a higher proportion of CD38- cells. Were overall engraftment levels of the MDS cells influenced by DAB2 knockdown, and what was the functional consequence of restricting TGFbeta activation? What is the impact of constitutive DAB2 expression on normal human CD34+ hematopoietic stem and progenitor cells?’

We analyzed engraftment levels in mice transplanted with del(5q) MDS-L cells, in which miR-143/145 are haplodeficient, and found that knockdown of *DAB2* significantly reduced the number of human CD33⁺ myeloid cells in the blood (new Fig 8j in the revised manuscript and p. 14, line 5) consistent with inhibition of myeloid progenitor expansion when the experiment was ended at 5 weeks (Fig. 8i). Despite several attempts and creating a novel construct to transduce CD34⁺ cord blood cells, we were not able to obtain increased expression of *DAB2* in human CD34⁺ cord blood cells. Given that the above data in a human del(5q) cell line confirms our findings in the miR-143/145-null mice, we feel that additional data with overexpression of *DAB2* in normal human cells would not provide substantial additional material that would support a link between miR-143/145 and *DAB2*.

‘6. Some of the claims made in the text are not appropriate and should be toned tone or revised. On page 6, the authors hypothesize that “HSC activity also appeared to be reduced in miR-143/145+/- mice..”. This hypothesis is based on CFU assay, which does not provide an HSC readout-in addition to a p value of 0.43 which is far from significant.’

We apologize that the reference to Supplementary Fig 1c was not correct, as the statement about HSC activity in heterozygote mice was based on the LDA data in

Supplementary Table 2. The difference between WT and miR-143/145^{+/-} was indeed not significant and we have revised the statement (p. 6, line 12).

‘On page 8, they claim that DAB2 did not affect progenitor activity in primary CFU assays, while Fig. 5a shows a statistically significant reduction.’

We thank Reviewer #2 for pointing out this discrepancy. The difference is indeed significant although the effect size is very small, and significant differences were not achieved in every experiment. The text in the Results section has now been corrected (p. 8, line 21).

‘On page 9 (line 9), the relative increase in GMPs within DAB2-GFP cells with respect to the YFP control cell compartment is not fully convincing, given that all progenitor compartments show a similar trend (with variability between mice), rather suggesting that the DAB2-GFP compartment shows a reduction in Lineage+ cells. It may be problematic to compare subpopulations within the rare GFP+ cell compartment (comprising on average <10% at 20 weeks) to the much bigger YFP compartment.’

We agree that it may be problematic to distinguish GFP⁺ progenitor populations given the differences in engraftment levels. Although the observed increase only reached significance in the GMP compartment ($P = 0.05$), we acknowledge that the amount of data is not sufficient to attribute the expansion of *DAB2* cells to one progenitor compartment. Therefore, we have toned down the statements in the Results (p. 9, line 4) and Discussion sections (p. 16, line 20).

We also thank Reviewer #2 for the alternative explanation regarding a potential reduction in Lin⁺ cells. We have analyzed the percentage of Lin⁺ cells in Vector and *DAB2* populations but did not find a difference ($P = 0.66$) (Response Fig. 4 below), again acknowledging the issues with differential engraftment levels.

Response Figure 4. Marrow analysis of competitive DAB2 transplant. Bone marrow was harvested at 20 weeks post-transplant. Cells were gated on the transduced population to determine the proportion of lineage-positive cells within each of the GFP or YFP compartments ($n = 8$).

‘On page 11, the authors say that “miR-143/145^{-/-} mice showed leukocytosis, thrombocytopenia and anemia”, while the group averages referred to in Fig.7a show WBC, HGB and PLT levels within the normal range, even in the the 143/145^{-/-} group.’

Since our original submission, we have analyzed data from more mice. Blood counts from a larger cohort of mice aged 80 weeks (WT n = 28, KO n = 23) showed a significant relative increase in leukocytes ($P = 0.03$) and decreased hemoglobin ($P = 0.05$) and platelets ($P = 0.05$) (new Fig. 7a in the revised manuscript). As the reviewer points out, miR-143/145^{-/-} group means are still within the normal range (Mazzaccara et al. PLoS 2008). Nevertheless, a larger proportion of miR-143/145^{-/-} mice displayed leukocytosis (WBC > $14 \times 10^3/\text{mm}^3$, 17% in miR-143/145^{-/-} vs 7% in WT). These data are described in the Results section on p. 11, line 28, and are consistent with the DAB2 experiments showing that only a proportion of mice develop myeloid malignancy.

Reviewer #3 (Remarks to the Author):

'Karson et al show that DAB2 is a target of micrnas deleted in 5q- MDS. They show that increased expression of DAB2 due to 5q deletion leads to activation of TGFb-smad signaling and leads to stem and progenitor defects in vivo.

The studies are comprehensive and elegantly done. The use of multiple mouse models support the conclusions. The findings provide a link between a genetic deletion and activation of TGF signaling in MDS and advance the field.'

'Some minor concerns:

1. The gapdh loading control in the western blots appears to be from a different membrane.'

We thank Reviewer #3 for pointing this out. We have checked the loading controls in all western blots and made changes where required.

'2. papers showing activation of smad2/3 in human mds can be cited for completion'

In the Discussion section (p. 15, line 17), we have now included three references, including the paper by Zhou et al. in Blood (2008) which specifically showed that the Smad2 protein is heavily phosphorylated in MDS bone marrow progenitors.

'3. List of all TGF pathway genes that were altered and are micrrna targets should be moved to main figures instead of supplementary'

We have moved the diagram from Supplementary Figure 2 in the manuscript to main Figure 2d in the revised manuscript. A table listing all del(5q) enriched miR143/145 targets has been added to the revised manuscript (new Supplementary Table 4).

REVIEWERS' COMMENTS:

Reviewer #1 (Remarks to the Author):

Overall, the authors have reasonably adequately addressed my concerns. The inclusion of more data has improved the manuscript. I still do find it a little odd that the authors wrote in the first version that 143/145-/- mice live shorter, whereas this is clearly not the case. Now they agree that the effect may be so subtle that it requires large numbers of mice to be analysed . They now write " Given that wild-type mice are also approaching the end of their lifespan, the number of animals and time required to power this question, we feel that this is beyond the scope of the current manuscript". In fact, I did not ask the authors to do life span studies. but just suggested that if there are no lifespan differences in the cohorts used, they should not write that 143/145-/- mice live shorter.

Reviewer #2 (Remarks to the Author):

The authors have sufficiently addressed my concerns.

RESPONSE TO REVIEWERS' COMMENTS:

Reviewer #1 (Remarks to the Author):

“Overall, the authors have reasonably adequately addressed my concerns. The inclusion of more data has improved the manuscript. I still do find it a little odd that the authors wrote in the first version that 143/145-/- mice live shorter, whereas this is clearly not the case. Now they agree that the effect may be so subtle that it requires large numbers of mice to be analysed . They now write " Given that wild-type mice are also approaching the end of their lifespan, the number of animals and time required to power this question, we feel that this is beyond the scope of the current manuscript". In fact, I did not ask the authors to do life span studies. but just suggested that if there are no lifespan differences in the cohorts used, they should not write that 143/145-/- mice live shorter.”

We have checked our manuscript to ensure that we do not claim that the miR-143/145-null mice exhibit shorter lifespans.

Reviewer #2 (Remarks to the Author):

“The authors have sufficiently addressed my concerns.”